# Millennial scale persistence of organic carbon bound to iron in Arctic marine sediments

Johan C. Faust[1✉], Allyson Tessin[2], Ben J. Fisher [1], Mark Zindorf [3], Sonia Papadaki[4], Katharine R. Hendry [4], Katherine A. Doyle[1] & Christian März[1]

Burial of organic material in marine sediments represents a dominant natural mechanism of long-term carbon sequestration globally, but critical aspects of this carbon sink remain unresolved. Investigation of surface sediments led to the proposition that on average 10-20% of sedimentary organic carbon is stabilised and physically protected against microbial degradation through binding to reactive metal (e.g. iron and manganese) oxides. Here we examine the long-term efficiency of this rusty carbon sink by analysing the chemical composition of sediments and pore waters from four locations in the Barents Sea. Our findings show that the carbon-iron coupling persists below the uppermost, oxygenated sediment layer over thousands of years. We further propose that authigenic coprecipitation is not the dominant factor of the carbon-iron bounding in these Arctic shelf sediments and that a substantial fraction of the organic carbon is already bound to reactive iron prior deposition on the seafloor.

[1] School of Earth and Environment, The University of Leeds, Leeds, UK. [2] Department of Geology, Kent State University, Kent, OH, USA. [3] Laboratoire Environnement Profond, Ifremer - Centre de Bretagne, Plouzané, France. [4] School of Earth Sciences, University of Bristol, Bristol, UK. ✉email: J.Faust@leeds.ac.uk

Organic carbon (OC) burial in shelf sediments plays an important role in the global carbon cycle as ~87% of the estimated $169 \times 10^6$ tons of OC deposited at the seafloor were buried in these shallow parts of the ocean each year[1]. However, the fundamental, physical, biological, and chemical processes that control OC preservation, including sedimentation rate[2,3], presence and absence of oxygen[4–6], selective preservation of biochemically unreactive compounds[7,8], and protection of organic matter through interactions with a mineral matrix[9–11] are complex and not fully understood. Even though a possible chemical association between iron and OC in soils were identified more than half a century ago[12], clay minerals were historically viewed as the primary controller for mineral hosted sedimentary carbon. The potentially global importance of reactive iron oxides ($Fe_R$; nanoparticulate and amorphous phases of ferric (oxyhydroxides) for OC preservation in marine sediments has only recently been recognised[13].

At the $Fe^{2+}/Fe^{3+}$ redox boundary, typically in the upper centimetres of a shelf sediment profile, oxidation and precipitation of upward diffusing $Fe^{2+}$ (liberated to the pore waters by dissimilatory iron reduction) leads to an enrichment of sedimentary $Fe_R$[14]. The OC has a strong affinity to these freshly precipitating Fe(III) phases (e.g., ferrihydrite) and the resultant iron/OC association, through coprecipitation of OC within or sorption to reactive iron phases is assumed to promote long-term stabilisation of sedimentary organic matter[13,15–17]. Thus, reactive iron phases may serve as an efficient shuttle to promote carbon burial as OC associated with these Fe(III) phases should be protected against microbial degradation, allowing it to bypass the efficient early diagenetic degradation regime[18] and to be buried into anoxic sediments, where the OC preservation potential is much higher.

To date, the mechanisms stabilising OC with $Fe_R$ in marine sediments have mainly been studied in surface sediments[13,19–23]. These studies show that the fraction of the total OC bound to $Fe_R$ (f$OC$-$Fe_R$) is on average 10–20%, with values ranging from ~0.5 to 40%. A series of factors, such as binding mechanisms of OC to $Fe_R$, sediment mineralogy, organic matter composition, and iron redox cycling were invoked to explain the wide variations of f$OC$-$Fe_R$. However, the term "surface sediment" is ill-defined and the depth below the seafloor of the investigated sediments is not consistent between different studies (ranging from 0.5 to 3 cm or even unspecified). Moreover, information about the position of studied sediment samples relative to the $Fe^{2+}/Fe^{3+}$ redox interface is usually absent. However, this information is critical, as Fe(III) phases making up the $Fe_R$ pool are highly redox-sensitive and under anoxic conditions deeper in the sediment, dissimilatory iron reduction[24] may affect the stability of the $OC$-$Fe_R$ bonding. The stabilisation of OC by $Fe_R$ may therefore be transient and only stabilise OC in the oxic surface sediment layer. Hence, while iron redox cycling has been proposed as a controlling factor of f$OC$-$Fe_R$, current findings based on surface sediment investigations might be biased by differences in the depth of oxygen penetration and the $Fe^{2+}/Fe^{3+}$ redox interface at the different locations. Moreover, downcore investigations of $OC$-$Fe_R$ will not only provide a better understanding of the role of early diagenesis in $OC$-$Fe_R$ generation and stability, they will also help to reveal the source of the $OC$-$Fe_R$ (allochthonous vs. autochthonous) and allow to identify the relative contributions of $OC$-$Fe_R$ that was formed on land during the transport process, or at the sediment-water interface.

Besides Fe(III) phases, Mn(III/IV) (oxyhydroxides) ($Mn_R$) also strongly interact with OC in marine sediments[25–27]. However, similar to the $OC$-$Fe_R$ coupling, $OC$-$Mn_R$ in marine sediments has so far only been investigated in surface sediments and a paucity of information remains on the abundance of carbon associated with manganese oxides and their potential role in stabilising OC over longer timescales. It is therefore unclear if manganese oxides help to transfer OC from the sediment surface carbon cycle to the geological carbon cycle or if $Mn_R$ plays a minor role in OC stabilisation compared to $Fe_R$[25,26].

To better understand the effect of sedimentary degradation processes on the formation and stability of the $OC$-$Fe_R$ and $OC$-$Mn_R$ association over long (millennial) timescales, we chemically analysed pore water and sediment samples retrieved at four coring sites along a south-north transect across the Arctic Barents Sea shelf area (Fig. 1). Iron and manganese (oxyhydroxide) reduction play an important role in organic matter degradation in this region[28,29] and it is therefore a suitable location to study the combined diagenetic fate of OC, iron, and manganese. Moreover, the Barents Sea region currently experiences the greatest warming in the Arctic, a dramatic loss of sea ice[30] and the highest increase of primary productivity across the Arctic ocean[31]. Thus, the transformation from an icy-land into an open ocean force the entire Barents Sea ecosystem to adapt and restructure, which affects the Arctic carbon cycle through changes in atmospheric $CO_2$ uptake, pelagic-benthic coupling, organic matter sedimentation, and long-term sequestration[32–37]. Understanding the mechanisms responsible for enhancing the stability and long-term storage of OC worldwide and especially in the Arctic is important for predicting how the global carbon cycle will respond to climate change.

Here, we examine (I) the persistence of f$OC$-$Fe_R$ below the oxygenated part of the sediment over millennial time scales, (II) the effect of the $Fe^{2+}/Fe^{3+}$ redox boundary on $OC$-$Fe_R$ binding mechanisms, (III) the importance of different Fe(III) phases for binding OC in competition with other chemical species such as arsenic, (IV) the role of manganese oxides in stabilising OC in marine sediments on longer timescales, and (V) potential allochthonous $OC$-$Fe_R$ contribution.

## Results and discussion

**Reactive and total iron sources.** Our results show that at all four study sites (B13–B16), the Fe/Al is highly correlated to the sedimentary $Fe_R$ contents ($r \geq 0.95$; Supplementary Fig. S1), the reactive iron fraction of total iron (f$Fe_R$) shows the same downcore pattern as $Fe_R$ and both parameters are closely related to the total sedimentary iron contents (Figs. 2, 3 and Supplementary Fig. S2). The close relationship between Fe/Al, total iron, and $Fe_R$ could lead to the conclusion of a common source, e.g. ref. [38], i.e., terrigenous influx from Svalbard (Fig. 1). However, as expected and revealed by the comparison of our $Fe^{2+}$ pore water and f$Fe_R$ profiles, early diagenesis plays an important role in the generation of the observed f$Fe_R$ patterns over depth (Fig. 2). At the $Fe^{2+}/Fe^{3+}$ redox boundary, oxidation and precipitation of upward diffusing $Fe^{2+}$ leads to an enrichment of Fe/Al and reactive Fe(III) phases, which results in an accompanying increase of f$Fe_R$ at all four sites. At stations B13, B14, and B16, the iron redox interface occurs in the top 5 cm at each site. At station B15, oxygen penetrates deeper into the sediment[39] and the iron redox interface is located between ~5 and 12 cm. Therefore, diagenetic processes have a strong impact on f$Fe_R$ patterns at the study location. Seasonal and annual primary productivity changes, for example, through the differences in sea ice cover during the three consecutive sampling years possibly caused variations in the depth of the redox boundary. However, early diagenesis variability had probably only a minor effect on our f$Fe_R$ results as the iron redox interface was remarkably stable in all investigated cores during our sampling campaigns in summer 2017, 2018, and 2019 (Supplementary Fig. S4). Moreover, the stable redox interface also indicates only minor disturbance of the sediment

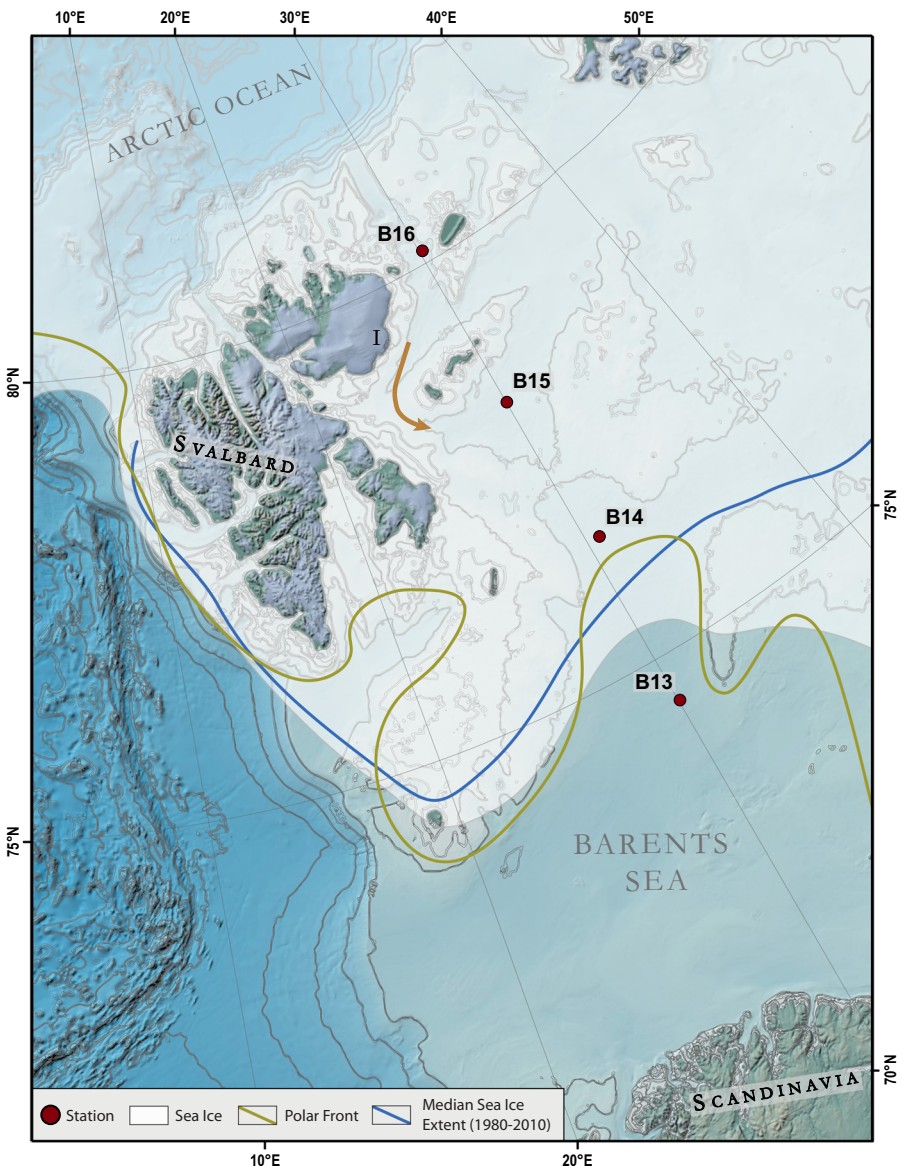

**Fig. 1 Map of the study area and sampling stations.** The Barents Sea is the largest pan-Arctic shelf sea covering an area of 1.6 million $km^2$ with an average water depth of 230 m[67]. There are several extensive overviews and reviews about the modern climate setting and ecosystem of the Barents Sea and we refer to these references for a detailed description of the physical and ecological conditions[34,68–72]. The general oceanic circulation pattern of the western Barents Sea is dominated by the relatively warm northward flowing North Atlantic Current which enters the Barents Sea from the south-west and the southward flowing cold Arctic currents entering the Barents Sea from the north-east. The relatively sharp boundary between these water masses forms the oceanographic Polar Front (golden line)[44], which is mainly determined by the bathymetry and is, therefore, relatively stable from year to year[73]. The northern Barents Sea is seasonally ice covered with maximum and minimum ice coverage in March–April and August–September, respectively. The heat content of the Atlantic water keeps the southern Barents Sea permanently ice-free. River runoff into the Barents Sea is limited. Rivers on Svalbard and in Norway are small and often drain into fjords. Thus, sediment discharge through river inflow is low and the main processes responsible for the Barents Sea surface sediment distribution are re-deposition by winnowing from shallow banks into troughs and depressions and deposition from sea ice. Hence, sedimentation rates are generally low, 0.04–2.1 mm/y since the last glacial period[48]. The brown arrow indicates the proposed transport of iron-rich sediments from Nordaustlandet (I) into the central Barents Sea. The map was created using the IBCAO V. 3.0 dataset[74].

column through bioturbation, which is in accordance with a recently reported very shallow mean bioturbation depth (<1 cm) at all investigated stations[40].

At stations B13, B14, and B16, we found no indication of significant changes in the external input of $Fe_R$ phases to the seafloor over time. Sediment cores from these sites show surface enrichments of $fFe_R$ with maxima (~20–30%) in the top 5 cm, which are related to the precipitation of authigenic Fe(III) phases and relatively stable values (~10%) to the core bottom (Fig. 2). In comparison to the other stations, $fFe_R$ at station B15 is much

higher and shows a distinct peak of up to 51% between 9 and 15 cm. This peak corresponds to high total iron contents of up to 8% and a reddish/pink, fine-grained sediment layer. Such reddish sediment layers are known in sediments north and west of Svalbard and probably originate from iron-rich Devonian sandstones in central Svalbard[41]. To our knowledge, no such pink/reddish sediment layers have previously been reported from the central Barents Sea. But modern glacial discharge at the eastern side of Svalbard (Fig. 1; Nordaustlanded) creates reddish meltwater plumes[42], and sea ice covered with reddish sediment

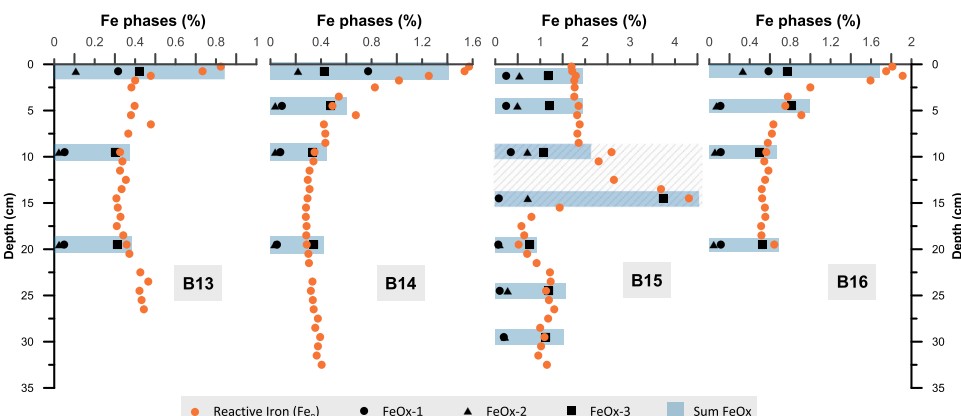

**Fig. 2 Downcore measurement of pore water iron content and sedimentary total iron, reactive iron, total organic carbon content, as well as the fraction of total organic carbon bound to reactive iron and the OC to iron molar ratio.** The pore water iron content represents the average iron concentration from three different cores at the same coring location. Black squares are uncalibrated radiocarbon dates in kyr BP. The shaded grey area at station B15 marks the location of a reddish/pink sediment layer.

**Fig. 3 Comparison between the downcore profiles of reactive iron (Fe$_R$)[13] and the results of the sequential iron extraction[75].** The three iron phases from the sequential iron extraction, shown by black circles, triangles and squares, are: Fe bound to carbonates or very labile iron oxides (FeOx-1); poorly crystalline Fe (oxyhydr)oxides like ferrihydrite (FeOx-2) and crystalline Fe (oxyhydr)oxides such as hematite and goethite (FeOx-3). Blue bars indicate the sum of FeOx1–3 which shows that the dithionite extraction of Fe$_R$ (orange dots) represents the entire reactive iron pool. The grey hatched area (B15) indicates a pink, iron-rich sediment layer, which is enriched in more crystalline iron phases (FeOx-3).

(Supplementary Fig. S3) has been observed during our and other expeditions around eastern Svalbard. The Barents Sea current system transports the fine-grained components incorporated in the meltwater plume south-westward and into the central Barents Sea (Fig. 1; refs. [43–45]), towards the location of station B15. Hence, we suggest that the high iron content at station B15 is related to north-eastern glacial sediment discharges on Nordaustlanded.

**Reactive iron content and OC-FeR bonding**. The large downcore variations in the $fFe_R$ profiles raise the question of whether the $Fe_R$ content has a substantial control on the fraction of OC bound to $Fe_R$. Due to the strong affinity of OC to reactive Fe(III) phases (e.g., ref. [46]), we would expect that an increase in $Fe_R$ results in an overall higher OC-$Fe_R$, presuming sufficient OC is present. However, in our downcore records and agreement with data from the beach and marine surface sediments[47,48], OC-$Fe_R$, as well as the fraction of total OC bound to $Fe_R$ (fOC-$Fe_R$), do not seem to be controlled by the amount of available sedimentary $Fe_R$. At stations B13, B14, and B16, fOC-$Fe_R$ shows a slight gradual decrease from the core top to the bottom and does not seem to follow the $fFe_R$ profile. And fOC-$Fe_R$ at station B15 shows principally no difference between the $fFe_R$-rich (up to 51%) reddish sediment layer at 9–15 cm depth and the sediments above (Fig. 2). This indicates that increased terrigenous iron and $Fe_R$ input does not necessarily result in higher fOC-$Fe_R$, and therefore, sedimentary iron and $Fe_R$ contents are not the exclusive controlling factors for the association of OC with $Fe_R$. A possible methodical explanation for this result, however, could be the extraction of iron phases with reduced surface reactivities, especially in very iron-rich sediment layers. To deconvolve the reactive iron species in more detail, we conducted a sequential iron extraction on selected samples from our sediment cores (Fig. 3), which reveals that the dominant reactive iron fraction in all sediment cores and in particular in the reddish sediment layer, is attributed to more crystalline iron oxide phases such as haematite and goethite (Fig. 3). The maturation and crystallisation of $Fe_R$ from more or less fresh phases like ferrihydrite to goethite/haematite decreases its surface area, reactivity, and solubility[49]. Thus, a large fraction of more crystalline oxides in the dithionite extractable $Fe_R$ pool may lead to an overestimation of $fFe_R$. Moreover, the gradual decrease of fOC-$Fe_R$ and OC-$Fe_R$ (Fig. 4) with increasing depth could be related to the predominance of more crystalline iron oxide phases in the $Fe_R$ pool below the surface sediments (Fig. 3). Thus, the loss of OC-$Fe_R$ may be caused by the maturation of these reactive iron phases and an accompanying release of the bound OC[50]. However, the OC-$Fe_R$ association not only protects the OC from degradation but is also believed to have a stabilising effect on the iron oxides and, therefore, helps to prevent the transformation to more crystalline phases, e.g. ref. [51]. Furthermore, OC associated with less reactive iron phases (e.g., goethite and haematite), probably via mono- or multi-layer sorption, is possibly more accessible for microbial degradation[52]. Further investigations are required to quantify the role of the different Fe(III) phases within the reactive iron pool in stabilising OC in natural sediments. Nevertheless, the decreasing trends of OC-$Fe_R$ and fOC-$Fe_R$ are accompanied by overall declining total sedimentary OC content with increasing depth at all stations (Fig. 4) and we cannot rule out that downcore variability in the fOC-$Fe_R$ has been affected by processes completely independent from iron. In fact, we fully acknowledge that the downcore patterns in the amounts of OC bound to $Fe_R$ may have been affected by various processes. These include the remineralisation of iron-bound OC over time, but also a combination of chemical, physical, and biological processes that affect sedimentary OC records, including

a variable fraction of OC being bound to clay minerals or variable amounts of non-bound OC being degraded[53,54]. Nonetheless, the fact that on average 19.2% of the total organic carbon remains bound to $Fe_R$ below the oxygenated surface sediment layer still highlights the important role that this OC-$Fe_R$ association plays in long-term carbon storage, despite the variance in environmental parameters over time. Moreover, as none of the presented iron metrics ($fFe_R$, $Fe_R$, Fe/Al) shows a consistent connection with either OC-$Fe_R$ or fOC-$Fe_R$ across all study sites our findings, reveal that the fOC-$Fe_R$ is not generally controlled by $Fe_R$ availability and a substantial increase of terrigenous iron and $Fe_R$ input does not necessarily have a direct effect on fOC-$Fe_R$. This raises the question of how much of the OC-$Fe_R$ is allochthonous, i.e., formed in the water column, in sea ice, on land, and how much is autochthonous, i.e., formed by biogeochemical processes within the sediments.

Suspended and bed sediments from the river and glacial systems show a clear link between OC and $Fe_R$[55], and substantial amounts of OC-$Fe_R$ in marine sediments may originate from land[19], especially at locations or during periods with high clastic sedimentation rates. The molar ratio of iron-bound OC to reactive iron (OC:Fe) has been interpreted as an indicator for the bonding mechanism between reactive iron and OC[13], with low values (<1) indicating the major OC-$Fe_R$ bonding form to be simple mono-layer sorption, while coprecipitation results in higher ratios[52]. As coprecipitation occurs at the oxic/anoxic interface in marine sediments, low OC:Fe ratios, for example, found in Mississippi river delta deposits were, therefore, interpreted as evidence for terrigenous OC-$Fe_R$ contribution[56]. Hence, low OC:Fe ratios at stations B15 and B16 could be interpreted such that sorption of OC is the dominant form of OC-$Fe_R$ association (Fig. 2). This suggests that at least part of the OC-$Fe_R$ in these sediments was formed on land, during the transport process, or at the sediment-water interface before burial[57]. This assumption is further supported by the high fOC-$Fe_R$ values (>10%) at the sediment surface at all four locations, above the $Fe^{2+}/Fe^{3+}$ redox boundary, where coprecipitation of OC with Fe(III) phases would occur. Such coprecipitation of OC and $Fe_R$ at the $Fe^{2+}/Fe^{3+}$ redox boundary has been proposed as the dominant mechanism behind the formation of OC-$Fe_R$ e.g. (refs. [13,17]). Our comparison of iron pore water profiles, $Fe_R$ contents, and fOC-$Fe_R$ provides the first real indication that coprecipitation at the iron redox boundary in marine sediments is less important for OC-$Fe_R$ bonding than previously assumed[13,17] and other controlling factors seem to be more important for the sedimentary fOC-$Fe_R$ contents.

Based on pore water data and the molar ratio of OC:Fe as a proxy for mono-layer sorption vs. coprecipitation, our data could indicate that at B14 and B15, coprecipitation may occur below the zone of $Fe_R$ precipitation (Fig. 2) and while OC:Fe at station B14 increases with increasing fOC-$Fe_R$, e.g., due to coprecipitation or multi-layer sorption, fOC-$Fe_R$ concentration at B15 decreases with increasing OC:Fe ratios. Even though OC:Fe and fOC-$Fe_R$ show similar trends at station B16, an increase in OC:Fe at the $Fe^{2+}/Fe^{3+}$ redox boundary does not result in higher fOC-$Fe_R$. Sediments from stations B14 and B13 show much higher OC:Fe values compared to B15 and B16, and fOC-$Fe_R$ and OC:Fe are clearly related in these two cores. It thus emerges from our data that the processes generating the OC-$Fe_R$ coupling at the different study sites are not exclusively related to the coprecipitation of OC and $Fe_R$ at the $Fe^{2+}/Fe^{3+}$ redox boundary. Several factors may be responsible for the different relationship between OC:Fe and fOC-$Fe_R$ at stations B13/14 vs. B15/16. For example, experimental laboratory studies showed that the organic matter composition can influence the OC:Fe ratio regardless of the adsorption and coprecipitation bonding mechanism[15,58,59]. Moreover, as our

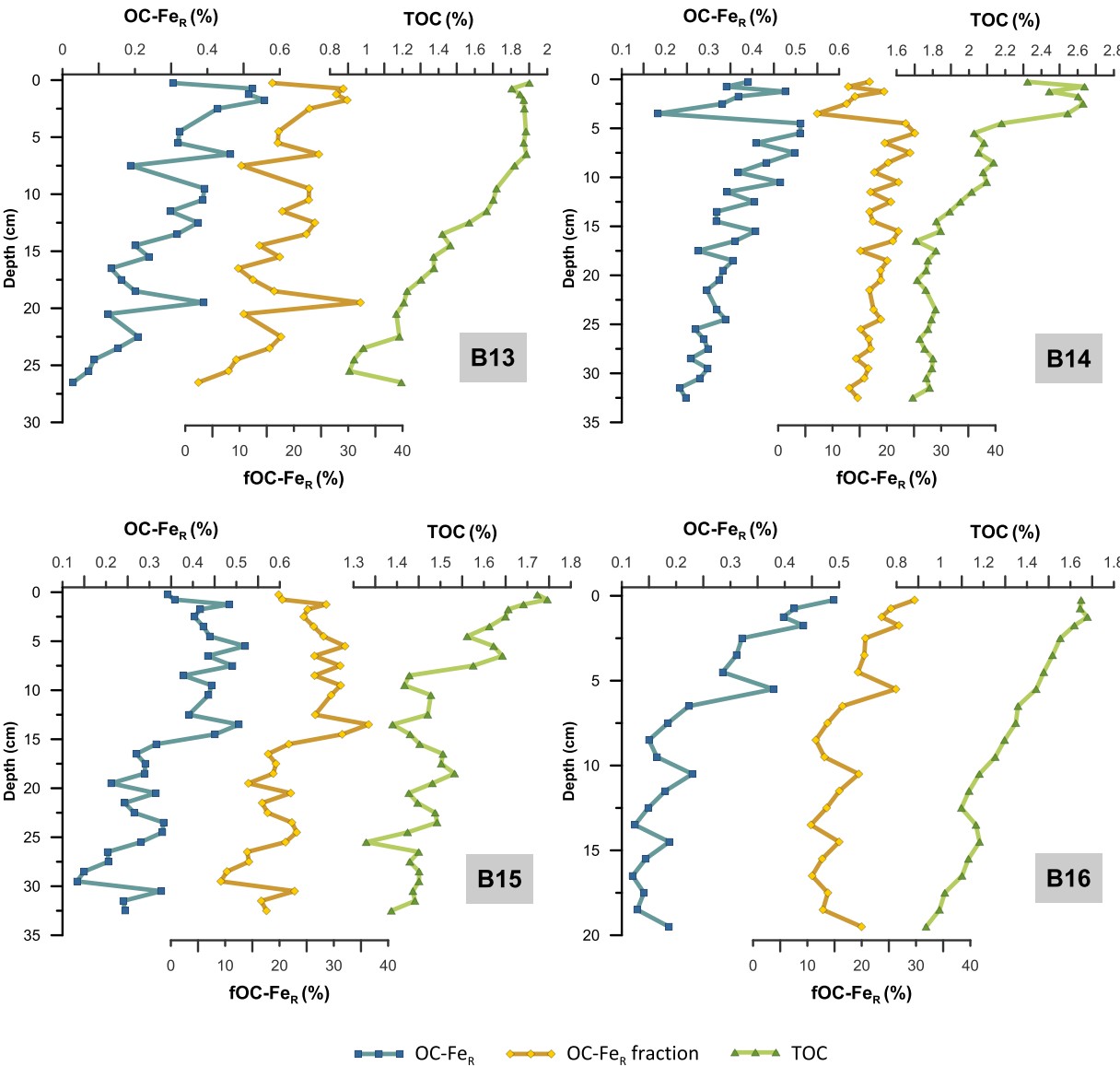

**Fig. 4 Downcore profiles of total OC content bound to reactive iron (OC-Fe$_R$), the fraction of total organic carbon bound to reactive iron (fOC-Fe$_R$), and total organic carbon content (TOC).** The gradual downcore decrease in all three parameters may indicate that sedimentary OC degradation also affects the OC bound to Fe$_R$, even though fOC-Fe$_R$ values are still relatively high in the lower part of these Arctic sediment cores. Other factors than diagenetic processes, such as environmental change, OC, and allochthonous OC-Fe$_R$ input changes, probably also play a role during the time span investigated here. Note the different scale of the x-axis (sediment core depth) and y-axis (TOC) between each core.

sediment cores represent a time span of several thousands of years, microbial degradation might have, over time, selectively modified the adsorbed or coprecipitated OC-Fe$_R$ content[60,61]. Additionally, the determination of Fe$_R$ via chemical extraction yields an operationally defined reactive iron pool, including Fe$_R$ that is not associated with OC. The molar ratio of OC:Fe might, therefore, be biased and especially low OC:Fe ratios, as in core B15 and B16, should be interpreted with care[62]. Finally, crystalline iron (oxyhydroxides) in natural sediments are impure, which influences the dynamics of the OC to Fe$_R$ association. In a competition to OC, phosphate, arsenic, and other heavy metals have a strong affinity to iron (oxyhydroxide) surfaces and can, therefore, influence the OC:Fe ratio as well as fOC-Fe$_R$[16]. Arsenic contents in the Barents Sea surface sediments are strongly related to Fe$_R$ contents ($r = 0.9$)[48]. Our downcore data show similar results with a clear relationship between As/Al, Fe/Al, and Fe$_R$, especially at the Fe$^{2+}$/Fe$^{3+}$ redox boundary (Supplementary

Fig. S5). Hence, we speculate that the strong relationship between arsenic and Fe$_R$ indicates that arsenic sorption could affect the mineral surface properties and reactivities of the Fe(III) phases and, therefore, their capacity to bind to OC[16].

**The role of reactive manganese in OC stabilisation.** Manganese is another element that potentially interacts with OC in marine sediments and might have biased existing fOC-Fe$_R$ estimates[23,26,27]. Besides Fe(III) phases, Mn(III/IV) (oxyhydroxides) (Mn$_R$) also strongly interact with OC in marine sediments[25], but their effect on carbon stabilisation in natural sediments is almost completely unconstrained. In marine sediments, Mn$_R$ is typically reductively dissolved below the depth of oxygen penetration[14]. Thus, the OC-Mn$_R$ association might not be stable over longer timescales due to the strong effect of the oxic-anoxic redox interface on Mn$_R$ stability[27]. Our data from the Barents Sea show that Mn$_R$ is strongly related to total manganese

contents and a rapid increase of pore water $Mn^{2+}$ is accompanied by a decrease in $Mn_R$ and total manganese close to the sediment surface. Below the top few cm of our cores, $Mn_R$ is virtually absent at station B14 and very low at stations B13 (<38 ppm) and B16 (<55 ppm; Supplementary Table S3 and Fig. S6). Therefore, we propose that $Mn_R$ in marine sediments is not important or plays only a very minor role, in the stabilisation of OC on longer timescales.

**Synthesis and implications**. Based on our investigation of the effect of sedimentary degradation processes on the formation and stability of the OC-$Fe_R$ association in the Arctic marine sediments, we posit that an increased influx of iron or a higher fraction of $Fe_R$ does not necessarily enhance fOC-$Fe_R$ values and that iron redox cycling and associated authigenic $Fe_R$ formation are less important for the stabilisation of OC in marine sediments than so far assumed, e.g. (refs. [17,23,46,57]). We show that significant amounts (>10%) of fOC-$Fe_R$ are present above the iron redox zone, before OC-$Fe_R$ coprecipitation presumably occurs, which suggests that at least some of the OC-$Fe_R$ binding forms on land, during transport to the seafloor, or at the sediment-water interface prior to deposition. Moreover, the association between OC and $Fe_R$ via coprecipitation does not necessarily result in higher fOC-$Fe_R$ and is, therefore, not the dominant factor controlling fOC-$Fe_R$ in the Barents Sea sediments. Other factors such as organic matter composition[15,58,59], inconsistent effects of different binding mechanisms on organic matter loadings[60,61], and changes in Fe(III) phase reactivity due to incorporation and adsorption of other elements with a high affinity to $Fe_R$[16], likely all play a (combined) role in natural environments.

A recent investigation of the Barents sea surface sediment samples found that the spatial distribution of the fOC-$Fe_R$ content seems to be unrelated to sea ice cover, Atlantic water inflow, proximity to land, grain size distribution, or sediment composition[48]. Although more work is needed to elucidate the impact of climate and environmental changes on the fOC-$Fe_R$ in marine sediments, the finding of this study could indicate that future Arctic warming might neither enhance nor decrease average carbon burial through the adsorption to iron oxides as, even though fOC-$Fe_R$ profiles at all stations show some degree of variability, total fOC-$Fe_R$ values averaged over all depths of all four sediment cores are surprisingly similar (B13: 18.1 ± 7.3%, B14: 17.7 ± 3.6%, B15: 22.5 ± 6.4%, B16: 17.9 ± 5.6% (mean ± s.d.)). These values are in agreement with published estimates for the Barents Sea (21.0 ± 8.3%) as well as global marine surface sediments (21.8 ± 8.6%)[13,48]. Thus, while highlighting new and important complexities in the coupling of OC with iron in marine sediments, our results clearly underline the importance of reactive iron phases for OC burial. In particular, we show that despite clear evidence for the iron reduction in the studied deposits, on average 19.2% of the total OC remains bound to $Fe_R$ below the oxygenated surface sediment layer over thousands of years in the Arctic marine sediments. This shows that the rusty carbon sink is not disabled by diagenetic processes affecting both OC and Fe(III) phases and data from surface sediments can be used for meaningful estimates of OC-$Fe_R$ burial rates.

## Methods

**Sediment cores: sampling and preparation**. In July 2017, 2018, and 2019, sediment cores were collected by using a multi-corer at the same four stations (Supplementary Table S1) along a south-north gradient in the western Barents Sea (Fig. 1). One tube per multi-corer deployment at each station visited in 2017 was sliced at 0.5 cm intervals until 2 cm core depth, and at 1 cm intervals thereafter. Samples were stored in plastic bags at –20 °C immediately after recovery on-board the Royal Research Ship James Clark Ross. Prior to any chemical sediment analysis, all samples were freeze-dried and homogenised by gentle grinding using an agate

mortar and pestle. For pore water analysis, four sediment cores were collected in all three years from three (two at B14 in 2017) sequential multi-corer deployments at each station (including the coring site for sediment sampling), with about 20–50 m distance between each deployment to account for spatial variability in sediment properties. Pore water was extracted with Rhizon samplers[63] inserted into pre-drilled Perspex core liners ($D$ = 110 mm). Following extraction, pore water samples from the same sediment depths from four core tubes per multi-corer deployment were combined into acid-washed and MilliQ-rinsed vials to reach maximum pore water volumes for individual sediment layers. Pore water splits of 3 mL for cation analysis were acidified with 10 µL concentrated ROMIL-UpA™ HCl and stored at 4 °C.

**Sediment and pore water analysis**. Sedimentary bulk iron, manganese, aluminium, and arsenic contents were determined by wavelength dispersive X-ray fluorescence (XRF; Supplementary Table S4). A sample split of 700 mg was mixed with 4200 mg di-lithiumtetraborate ($Li_2B_4O_7$, Spectromelt A10), preoxidized at 500 °C with 1.0 g $NH_4NO_3$ (p.a.), and fused to homogenous glass beads. The glass beads were analysed using a Philips PW-2400 WD-XRF spectrometer calibrated with 53 geostandards at the University of Oldenburg. Analytical precision and accuracy were better than 5% as checked by in-house and international reference materials. Pore water concentrations of iron and manganese were determined by inductively coupled plasma optical emission spectrometry (Thermo Scientific iCAP 7400 Radial ICP-OES) at the University of Leeds. Analytical precision was ±3.5% and results were provided in the Supplementary Table S2.

**OC, iron and manganese extraction, and analysis**. To quantify the amount of OC bound to iron and manganese (oxyhydroxides) in our samples, we applied a method described in detail by Lalonde et al.[13] and Salvadó et al.[19]. Briefly, 0.25 g of each sample was transferred into 30 ml centrifuge tubes. Fifteen millilitres of a solution containing 0.27 M trisodium citrate ($Na_3C_6H_5O_7$·$H_2O$) and 0.11 M sodium bicarbonate ($NaHCO_3$) was added, mixed and heated up to 80 °C in a water bath. 0.1 M sodium dithionite ($Na_2S_2O_4$; 0.25 g) was added to the mixture, the temperature was maintained at 80 °C, and the tube was shaken every five minutes. After 15 min, the mixture was centrifuged for 10 min at 3360 $g$, the supernatant was decanted, and 200 µl of HCl were added to prevent Fe(III) precipitation. The remaining sediment samples were rinsed three times with artificial seawater and then freeze-dried. To quantify the OC loss unrelated to metal oxide dissolution, a control experiment was conducted: A 0.25 g aliquot of each sample was treated the same way as for the reduction experiment, but the complexing and reducing agents (sodium citrate and sodium dithionite) were replaced with sodium chloride to reach a solution of the same ionic strength. All samples were weighed after the experiment to account for mass loss. Dissolved iron and manganese in the supernatant and rinse water of the control and reduction experiments were analysed using a Thermo Scientific iCE3000 Atomic Absorption Spectrometer (AAS) at the University of Leeds, UK. Results are shown in the Supplementary Table S3 and the relative error of the analysis, based on eleven sample replicates, was ±3.2% for iron and 9.9% for manganese. Unlike iron, the dithionite extraction method is not well established for manganese. As manganese concentrations in all control experiments were below the detection limit, we assume that dithionite extracted manganese represents only reactive manganese phases[64].

We also performed a three-step sequential iron extraction on a subset of sediment samples ($n$ = 18) following the procedure of Poulton and Canfield[65], and März et al.[66]. The extracted iron fractions were operationally defined but usually constitute Fe bound to carbonates or very labile iron (oxyhydroxides) (FeOx-1), poorly crystalline Fe (oxyhydroxides) like ferrihydrite (FeOx-2) and crystalline Fe oxides such as haematite and goethite (FeOx-3). FeOx-1 was extracted using 1 M Na-acetate solution (pH 4.5) for 24 h; for FeOx-2 a solution comprised of 50 g Na-citrate, 50 g Na-bicarbonate, and 20 g ascorbic acid (per litre) at pH 8 for 24 h was used; Fe-S3 was extracted in a solution of 0.2 M Na-citrate and 0.28 M Na-dithionite buffered to pH 4.8 with 0.35 M acetic acid for 2 h. To ensure reproducibility, one sample was extracted in triplicate at all three stages, and the relative error was <2.9%. The iron fractions FeOx-1-3 were analysed using a Thermo Scientific iCE3000 Atomic Absorption Spectrometer (AAS) at Leeds University, UK.

The OC content of the bulk sediment before and after the reduction and control experiments was analysed on decarbonated samples using 10% (vol.) HCl, rinsed three times and dried overnight at 50 °C. OC content was determined with a LECO SC-144DR combustion analyser at the University of Leeds, UK (Supplementary Table S3). The certified reference material LECO 502-062 and blanks were included in every batch, and results are given in weight percentage. The relative error of the OC analysis was ±1.7%. To account for the mass loss during the extraction experiment we applied the mass balance calculation of Salvadó et al.[19] (Supplementary information). Note that, liquid-HCl decarbonation of the bulk sediment samples may also dissolve reactive iron phases in addition to carbonates. This could potentially liberate some iron-bound OC, which would bias our bulk OC results to lower values and thus bias our OC-$Fe_R$ results upwards (Supplementary information).

**Chronology**. Nine radiocarbon ages (Fig. 2) were obtained from benthic for-aminifera using a Mini Carbon Dating System (MICADAS) at the Bristol Radio-carbon Accelerator Mass Spectrometry facility. Uncalibrated carbon-14 ages are shown in the Supplementary Table S3.

## Data availability

The authors declare that all data supporting the findings of this study are available within the Supplementary Data.

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

## Acknowledgements
We thank the crew of the RRS James Clark Ross for their professional support during our expedition. Further, we would like to express our gratitude to Andy Connelly, Andrew Hobson, Fiona Keay, Carola Lehners, Corinna Mori, and Bernhard Schnetger for their help with the laboratory work at the University of Leeds and at the ICBM Oldenburg, as well as Heather Birch and Timothy Knowles for training in foraminiferal taxonomy and MICADAS analyses at the University of Bristol. This work resulted from the ChAOS project (NE/P006493/1), part of the Changing Arctic Ocean programme, jointly funded by the UKRI Natural Environment Research Council (NERC) and the German Federal Ministry of Education and Research (BMBF).

## Author contributions
J.C.F. was the lead author, wrote the manuscript, and created all figures. J.C.F., M.Z., A.T., K.D., and C.M. conducted fieldwork/sampling together and compiled datasets. J.C.F., S. P., B.F., and K.H. conducted all the required analytical work. All authors contributed early ideas, revised the initial manuscript and provided a lively discussion.

## Competing interests
The authors declare no competing interests.
