## [Peer Review File · Nature Communications]

REVIEWER COMMENTS

Reviewer #1 (Remarks to the Author):

Faust and colleagues quantify the percentage of organic carbon (OC) bound to reactive iron (FeR) and the influence of several variables such as iron source, Fe redox cycling, and OC loading on this OC-Fe interaction at four sites over three years in the Barents Sea. This paper adds significantly to our understanding of the “rusty” carbon sink in a number of areas, including key questions regarding the stability of these organo-mineral interactions below the Fe redox boundary, the stability of these associations over millennia, the relationship between the operationally defined FeR phases and the Poulton and Canfield classifications, and the importance (high or low) of coprecipitation in stabilizing OC. This manuscript is of significant interest to the scientific community, particularly those examining the role of mineral association in the long-term sequestration of organic matter in sediments and soils, globally. I have minor comments and some suggestions for the authors’ consideration.

Comment for consideration by authors: The authors have shown that OC is bound to the operationally defined reactive Fe at all depths in the sediment corresponding to thousands of years. This is interpreted as evidence that the OC is shielded from degradation for millennial timescales. With the exception core B15, the fraction of OC bound to iron (fOC-FeR) decreases with increasing depth (and conversely over time). From a simple reactivity point of view, this trend could be interpreted as evidence that OC-Fe is a relatively more reactive pool than the bulk OC. I do not believe this subtracts value from the findings and conclusions of this paper, but I do think some mention of this needs to occur in the discussion, particularly with reference to other mechanisms of carbon preservation in sediments. It seems that reactive iron is an important sink of OC for at least millennia, but if the decreasing fOC-FeR continues with depth then it does not represent the ultimate carbon sink.

Specific comments:

Lines 114-119: In the “Implications and Conclusions”, please add a sentence or two about how your findings provide insight into how the Fe-OC carbon sink is responding to these rapid changes in the Barents Sea and greater Arctic Ocean.

Line 184: “usual” should be “usually”

Line 228: Should “fFeR” be “porewater Fe” here since you do not have data for three consecutive years of FeR data?

Reviewer #2 (Remarks to the Author):

Synopsis

This manuscript describes the down-core interactions between organic carbon (OC) and reactive iron oxides (FeR) at four coring sites located near Svalbard in the Barents Sea / Arctic Ocean. From my reading, it seems the primary goal of this study was to answer the question: do OC-FeR interactions persist into deeper, more reducing portions of the sediment column, or is this OC re-released (and potentially re-exposed to remineralization) upon reduction and dissolution of FeR mineral phases? Answering this question is of critical importance for assessing the longterm role of the “rusty carbon sink” within the geologic carbon cycle; while previous studies on this topic have emphasized the total abundance of OC-FeR interactions in surface sediments, few (if any) have tested this specific question. As such, I find the topic of this manuscript to be well-suited for a broad journal such as *Nature Communications*.

By making a suite of solid-phase (wt. % Fe, OC, FeR, OC-FeR, Al, As, and Mn) and liquid-phase (porewater Fe²⁺ concentrations) porewater measurements, the authors come to two main conclusions: (1) a significant fraction of OC-FeR at these sites is delivered by allochthonous sources rather than being precipitated *in situ* in sediments, and (2) this OC-FeR does not rapidly diminish below the Fe(II)/Fe(III) redox transition and thus these interactions persist for ~millennia. The data are compelling, thorough, and of high quality (although I note that some of these measurements, particularly the iron speciation, are outside of my field of expertise and I

thus cannot comment too thoroughly on the methodology). I find these main conclusions to be well-articulated and relatively well-justified by the data.

Still, there are a few points and arguments that I think could be refined, and there are some minor conclusions and statements that I feel are not well supported by the data. Additionally, I suggest that the authors slightly adjust some of the ways in which they present the data in order to better emphasize these results. Below, I articulate these points in detail, followed by a list of minor, line-item comments. Once these changes have been made, then I support publication of the manuscript in *Nature Communications*. Please do not hesitate to contact me with any questions regarding this review.

Sincerely,

Jordon Hemingway

+1 760 445-3714

jordon_hemingway@fas.harvard.edu

“Larger” points

Use of relative, rather than absolute, OC-Fer data. Throughout the manuscript, the authors use the relative stability of the variable fOC-Fer (i.e., the *fraction* of OC that is bound to reactive iron) as their main argument for the persistence of these interactions. However, this stability simple shows that OC-Fer decays *at roughly the same rate as bulk OC*, particularly for the sites that show a relatively monotonic decrease in %OC with depth (i.e., B13, B14, and B16). When

plotting the weight % OC-Fer rather than fraction of total OC, the data tell a slightly different story---in all cores, there appears to be a clear (albeit noisy) decrease with depth. To me, *this* is the trend that really matters, since it speaks to the total OC preservation flux, rather than relative proportions. I point to specific instances where I think this slight shift in focus would improve the clarity of arguments in my line-item comments, below.

Additionally, the observation that iron-bound OC decays at roughly the same rate as bulk OC is interesting and should be explored more. Can the authors speculate as to some mechanistic reasons why this might be the case? (potentially using their Fe speciation data shown in Fig. 3).

Omission of pore-water O₂ data. Throughout the manuscript, the authors discuss the possibility of Fer precipitation below the oxygenated surface sediments. While the Fe²⁺ profiles are a useful redox indicator, I think it would be very helpful to also include pore-water O₂ profiles (I'm sure these data exist, no?) The sentence on L293-295 makes me also wonder if bioturbation could be influencing the Fer content of surface sediments; having pore-water O₂ information could also help speculate on the role (or lack thereof) of bioturbation.

Use of liquid-decarbonated %OC data. I would like to see a bit of discussion on how the authors think the liquid acidification procedure will bias %OC and, particularly, OC-Fer results. The carbonate content of these sediments was not reported, but presumably if there are considerable carbonates then %OC would be biased upward relative to unacidified samples. On the other hand, some iron phases (e.g., FeS) are known to be dissolved by HCl, which could remove some %OC and thus bias values downward relative to unacidified samples. Was DOC or Fe concentration of the supernatant measured? I suspect the authors could spend a few lines to come to a logical conclusion as to whether the reported values should be treated as minimum estimates, maximum estimates, or neither.

Line-item comments

L16: Change “the dominant mechanism” to “...a dominant mechanism” since carbonate burial is also quite important.

L23 (and 127-128): The mention of manganese here seems rather out-of-the-blue and given without context. I would appreciate a sentence or two in the abstract (and, especially, in the introduction) that articulates why Mn should also be included in this study (e.g., what are the open questions? How does this relate to OC-Fe? What has been studied in the past in this regard?) The majority of the manuscript (including the title!) deals specifically with iron, so placing Mn oxides within this context will better justify to the reader why these data are included here.

L43: I'm not sure if Berner 1970 is an appropriate reference here---this paper deals with OC as substrate for sulfate reducers, which leads to sulfide production and eventual pyrite precipitation. While, *technically*, this is a “association between iron and OC”, I don't think it quite supports what the authors are articulating here.

L47: I would recommend changing to “...oxidation and precipitation of upward...”

L52-55: The way this paragraph is set up, the authors are *specifically* discussing OC-Fe_R that has precipitated at the Fe(II)/Fe(III) redox boundary. This OC has, by definition, already “bypass[ed] the efficient oxic degradation regime...”, given that this redox boundary sits below the oxic zone. I suggest the authors rephrase this paragraph slightly to encompass allochthonous OC-Fe_R, which should indeed behave in the way the authors describe here.

L124: Change “persistency” to “persistence”

L126-128: See my above comment about contextualizing the introduction of manganese here.

L141: Remove the comma after “samplers”

L161: Remove the commas before and after “Salvadó”

L218: “...oxidation and precipitation of upward diffusing...”

L219: this also results in an accompanying increase of Fe/Al, which should be noted.

L256-257: I don’t think this sentence is strictly true. One would predict that an increase in *weight % Fe_R* would lead to an increase in *weight % OC-Fe_R*, but wouldn’t necessarily lead to a higher proportion of total OC bound to iron.

L259-261: I think this sentence is overstated. When I plot fFe_R vs. fOC-Fe_R, I see a pretty strong correlation ($R^2 = 0.7$) at site B16. But, again, I think it’s the *weight % Fe_R* vs. *weight % OC-Fe_R* correlation that is equally, if not more, important. Interestingly, when I plot these variables against one other, I see a relative strong correlation at sites B15 ($R^2 = 0.48$) and B16 ($R^2 = 0.85$), but not at sites B13 and B14. This points to some nuanced differences between sites that I think could be explored and discussed more.

Fig 3: I think there is an error in either Fig. 3 or Table S3---FeOx-2 and FeOx-3 (called Fe-S2 and Fe-S3 in Table S3 for some reason?) appear to be flipped.

L312-313: Again, I don’t think this sentence is strictly true. Is the increasing OC:Fe ratio deeper in the sediments at sites B14 and B15 *really* evidence for coprecipitation? Alternatively, this

could just be subtle differences in the rate of decrease of iron and OC with depth. When I look at reactive iron weight % with depth, it appears to be dropping across the two regions where OC:Fe is increasing (particularly at ~15cm at site B15, i.e., at the base of the red layer). To me, this argues against co-precipitation but rather suggests OC decreases with depth at a slower rate than Fe in this region. This should be discussed in more detail.

L348: Change “preferentially” to “preferential”

L382: Change “proves” to “indicates” or similar; absolute certainty is always a recipe for disaster!

Reviewer #3 (Remarks to the Author):

Dear Faust et al.,

Your work “Millennial scale persistence of organic carbon bound to iron in Arctic marine sediments” shows high-resolution spatiotemporal trends of organic carbon and iron in marine surface sediments across the ubiquitously important redox gradient. Coarse chronologies for these cores support the idea that Fe-OC associations persist across this gradient in the surface sediments, with (1) one component of Fe-OC associations being of allochthonous origin (approximately 10% of OC) and (2) another component of OC stabilized by authigenic precipitation of Fe-oxides within the sediment column, summing up to a total of around 20% OC associated with and stabilized by Fe-oxides. This work is an important steppingstone for understanding the mechanisms of the coupling between Fe and the global carbon cycle and presents high quality data borne from the fruit of exemplary study design and execution.

In the following I summarize my major comments and aim to stimulate additional discussion:

The discussion on the role Fe-oxides comes across as one sided as there is a lack of discussion on the role of clay minerals and the surface area that they provide for stabilizing sedimentary organic carbon (e.g., Keil et al., 1994; Mayer et al., 1994). What do the early diagenetic TOC decreases mean for downcore organic carbon - bulk mineral (surface area) relationships? I would find the mineral surface area discussion of great value as do Fe-oxides stabilize a disproportionate amount of OC relative to silicates and other mineral classes based on their abundance? I do find it remarkable that reactive Fe, which comprises less than 2 wt.% in the “normal” background sediments, should participate in stabilizing 20% of the OC, while silicates which are likely present in double digit quantities (?) would only be stabilizing part of the remainder of sedimentary OC. What kind of specific surface areas can we expect from these reactive Fe-phases? Can their theoretical mineral surface areas explain the disproportionate amount of OC stabilized by them (perhaps there is measured surface area data available)? At the same time, why is the amount of OC stabilized by Fe-oxides insensitive to the amount of reactive Fe available (e.g., lines 257-259; lines 276-277)? In some way, this contradicts OC-MSA relationships (e.g., Keil et al., 1994; Mayer et al., 1994) seen for bulk sediment. In an additional twist to this discussion, what are the relationships between silicates and Fe-oxides and are we able to tease apart OC associations between one or the other? See for example figure 2 in Kleber et al. (2007).

How was f_{OC-Fe_R} calculated? The methods in this manuscript mainly provide a distillation of the wet chemical approach. Adding explicit equations in a supplementary file would be useful (beyond referencing Lalonde et al., 2012, which are light on these details). Additionally, reporting on uncertainties for these estimates (like Lalonde et al.) would also be informative. For the sake of formality, adding equations for the other calculated parameters (e.g., Fe_R) would also be useful to see how blanks were subtracted, etc.

Precipitation and coprecipitation of organic carbon (with iron oxides) is often discussed in the manuscript and here I wonder what does this mean? With “precipitation” a process of transitioning from dissolved to solid phase is suggested/implied. Is dissolved organic matter removed from porewater and coprecipitated on authigenic Fe-oxides? Does dissolved organic matter solubility change across oxic-anoxic transition zones? Is there enough pore water dissolved organic matter present to support such process (e.g., lines 50, 296, 314)?

A local discussion which I think is worth having concerns Svalbard, which acts as a point source of petrogenic organic carbon (reworked kerogen) to the adjacent sediments (Kim et al., 2011). The recalcitrance of petrogenic organic carbon and its behavior is quite different from that of freshly synthesized organic carbon (e.g., from soils or from the ocean) and its associations with minerals (e.g., Blattmann et al., 2019). What role could rock-derived Fe and kerogen be playing in these sediments? Could they be one reason for the insensitivity of reactive Fe abundance to Fe-OC?

The speculative discussions involving manganese (minor element with much less than 1 percent abundance) and arsenic (trace element) seem too long for this type of article. Why should such minor and trace elements have a detectable effect (using the bulk and semi-bulk methods used in this study) on sedimentary OC stabilization? Beyond competing with OC sorption, multivalent ions can also be involved in cation-bridging (Keil and Mayer, 2014), so in my opinion the effect could also go the other way. I suggest keeping a hypothesis-driven focus for the discussion.

Finally, how are the conclusions to be placed in the global context with so much variability as reported by Lalonde et al., 2012? In this regard, I think it would be useful to contextualize and include (wherever possible) (estimates of) parameters in the discussion such as mixed layer depth, sedimentation rate, Fe and Mn porewater (e.g., perhaps combine into one figure in supplemental like Froehlich et al., 1979), and oxygen exposure time of organic carbon (e.g., Hartnett et al., 1998).

Here, I list a few minor comments:

Lines 114-122: This goal, while interesting and of course highly relevant for Earth Science, in my opinion, somewhat detracts from the main scope and message that the title and abstract outlines. I would suggest moving this into the later discussion or “implications and conclusions” section. This is merely a suggestion.

Lines 123-124: I suggest deemphasizing/removing the “focus on the potential effect of variable iron fluxes...” as this effect is discussed extensively later and deconvolved from the effects of most pertinent interest here.

Line 43: Soil scientists were much earlier to recognize this and in my opinion is worth pointing out if such historical references are made. See review by Beutelspacher (1955).

Line 164: Remove “well”.

Line 184: Usually

Lines 194-199: Was organic carbon content corrected to weight loss of sample due to loss of acid-soluble minerals?

Line 208: The introduction of the acronym fOC-Fe_R is somewhat confusing; here, it is not clear how it is defined whether as an “effect of xx on yy” or as what I think it is.

Line 214: Lead

Fig. 2: I suggest uniform scaling and removing porewater Fe average (moving this to the supplemental) and simply indicating the oxic-anoxic transition zone with a colored band within the profile to make the figure less busy.

Line 271: "Oxide" should be singular.

Lines 271-273: Needs reference(s).

Line 300: Please add references for who previously assumed this.

Line 306: Don't the blue bars show the sum of Fe_R from the sequential extractions? Perhaps I am confused here.

Lines 309-311: This sentence seems redundant.

Lines 324-326: There is a problem in the logic of this sentence: how can coprecipitated reactive iron be more reactive towards microbial reduction (this is a process)?

Line 328: What does preferentially mean in this context? Preferentially in what way?

Line 332: What is meant with impure?

Line 338: Should be Figure S5.

Line 348: grammar: preferential

To wrap up, this study has been executed in a manner which is methodologically meticulous and brings it to the forefront of the discipline with thoughtful sampling and exemplary resolution (both temporal and spatial). A valuable contribution which goes after the big question left by Lalonde et al. (2012). With major revisions, I am confident that this manuscript will become much stronger than it already is.

Sincerely,

Thomas Blattmann

Yokosuka 04.09.2020

References:

Beutelspacher, H. (1955) Wechselwirkung zwischen anorganischen und organischen Kolloiden des Bodens. Zeitschrift für Pflanzenernährung, Düngung, Bodenkunde 69, 108-115.

Blattmann, T.M., Liu, Z., Zhang, Y., Zhao, Y., Haghypour, N., Montluçon, D.B., Plötze, M. and Eglinton, T.I. (2019) Mineralogical control on the fate of continentally derived organic matter in the ocean. Science 366, 742-745.

Froelich, P.N., Klinkhammer, G.P., Bender, M.L., Luedtke, N.A., Heath, G.R., Cullen, D., Dauphin, P., Hammond, D., Hartman, B. and Maynard, V. (1979) Early oxidation of organic matter in pelagic sediments of the eastern equatorial Atlantic: suboxic diagenesis. Geochimica et Cosmochimica Acta 43, 1075-1090.

- Hartnett, H.E., Keil, R.G., Hedges, J.I. and Devol, A.H. (1998) Influence of oxygen exposure time on organic carbon preservation in continental margin sediments. *Nature* 391, 572.
- Keil, R.G., Tsamakis, E., Fuh, C.B., Giddings, J.C. and Hedges, J.I. (1994) Mineralogical and textural controls on the organic composition of coastal marine sediments: Hydrodynamic separation using SPLITT-fractionation. *Geochimica et Cosmochimica Acta* 58, 879-893.
- Keil, R.G. and Mayer, L.M. (2014) Mineral matrices and organic matter, in: Holland, H.D., Turekian, K.K. (Eds.), *Treatise on Geochemistry*. Elsevier, Oxford, pp. 337-359.
- Kim, J.-H., Peterse, F., Willmott, V., Kristensen, D.K., Baas, M., Schouten, S. and Sinninghe Damsté, J.S. (2011) Large ancient organic matter contributions to Arctic marine sediments (Svalbard). *Limnology and Oceanography* 56, 1463-1474.
- Kleber, M., Sollins, P. and Sutton, R. (2007) A conceptual model of organo-mineral interactions in soils: self-assembly of organic molecular fragments into zonal structures on mineral surfaces. *Biogeochemistry* 85, 9-24.
- Lalonde, K., Mucci, A., Ouellet, A. and Gélinas, Y. (2012) Preservation of organic matter in sediments promoted by iron. *Nature* 483, 198-200.
- Mayer, L.M. (1994) Surface area control of organic carbon accumulation in continental shelf sediments. *Geochimica et Cosmochimica Acta* 58, 1271-1284.

Note: All line numbers in the reply sections relate to the revised manuscript.

Reviewer #1:

Minor Comments:

With the exception core B15, the fraction of OC bound to iron (fOC-FeR) decreases with increasing depth (and conversely over time). From a simple reactivity point of view, this trend could be interpreted as evidence that OC-Fe is a relatively more reactive pool than the bulk OC. I do not believe this subtracts value from the findings and conclusions of this paper, but I do think some mention of this needs to occur in the discussion, particularly with reference to other mechanisms of carbon preservation in sediments. It seems that reactive iron is an important sink of OC for at least millennia, but if the decreasing fOC-FeR continues with depth then it does not represent the ultimate carbon sink.

Reply: We agree that the OC-FeR decrease is an interesting observation in regard to discussion about the role of FeR as an effective mechanism for carbon burial. We modified the discussion and now provide some ideas and possibilities as to why we see this change in the OC and OC-Fe content. But we have to admit that without further investigations into the nature of the OC bound to FeR as well as a better knowledge of the exact Fe phases present this is difficult to answer. Moreover, factors other than diagenetic processes, such as environmental changes, sediment source changes and OC input changes, during the time span investigated here probably play a role. The fact remains that despite strong Fe reduction in the sediments (in all three summers), quite a substantial portion of OC-Fe found at the sediment surface survives this redox process.

We added the following sentences to the discussion part of the revised manuscript (line 146-162): "Moreover, the gradual decrease of fOC-FeR and OC-FeR (supplementary figure S7) with increasing depth could be related to the domination of more crystalline iron oxide phase of the FeR pool below the surface sediments (Fig. 3). Thus, the loss of OC-FeR may be caused by maturation of these reactive iron phases and an accompanied release of the bond OC40. However, the OC-FeR association not only protects the OC from degradation. The association is also believed to have a stabilising effect on the iron oxides and, therefore, helps to prevent the transformation to more crystalline phases e.g. 41. Furthermore, a large amount of the less reactive iron phases (e.g. goethite and hematite) are allochthonous and OC associated to these phases, probably via mono- or multi-layer sorption, is more accessible for microbial degradation. Further investigations are required to quantify the role of the different Fe(III) phases within the reactive iron pool in stabilising OC in natural sediments. Nevertheless, the decreasing trend of the OC-FeR content is accompanied by a decrease in the total OC content but this trend is much less pronounced in the fOC-FeR profiles. This indicates that, even though the total OC content decreases, a large fraction of total OC content is associated with FeR. It needs to be considered that factors other than diagenetic processes, such as environmental change and OC input changes, probably play a role during the time span investigated here."

Line 114 – 119: In the “Implications and Conclusions”, please add a sentence or two about how your findings provide insight into how the Fe-OC carbon sink is responding to these rapid changes in the Barents Sea and greater Arctic Ocean.

Reply: As suggested by reviewer 3, in the revised version of the manuscript we moved line 114-119 into the synthesis section (line 233-240). Unfortunately, the sedimentation rate of the investigated sediment cores is very low (roughly 5 cm per 1000 years, figure 2), thus we cannot “see” how the OC-Fe changed during the recent time. However, we now added the following sentences “A recent investigation of Barents sea surface sediment samples found that the spatial distribution of the fOC-Fe_R content seems to be unrelated to sea ice cover, Atlantic water inflow proximity to land, grain size distribution or sediment composition⁵. Although more work is needed to elucidate the impact of climate and environmental changes on the fOC-Fe_R in marine sediments, the finding of this study could indicate that future Arctic warming might neither enhance nor decrease average carbon burial through the adsorption to iron oxides as, even though fOC-Fe_R profiles at all stations show some degree of variability, total fOC-Fe_R values averaged over all depths of all four sediment cores are surprisingly similar (B13: 18.1±7.3%, B14: 17.7±3.6%, B15: 22.5±6.4%, B16: 17.9±5.6% (mean±s.d)).”

Line 228: Should “fFeR” be “porewater Fe” here since you do not have data for three consecutive years of FeR data?

Reply: We changed the sentence for clarification.

All other very minor issues and technical comments highlighted by reviewer 1 have been considered in the revised version.

Reviewer #2

Main comments

Throughout the manuscript, the authors use the relative stability of the variable fOC-FeR (i.e., the fraction of OC that is bound to reactive iron) as their main argument for the persistence of these interactions. However, this stability simple shows that OC-FeR decays at roughly the same rate as bulk OC, particularly for the sites that show a relatively monotonic decrease in %OC with depth (i.e., B13, B14, and B16). When plotting the weight % OC-FeR rather than fraction of total OC, the data tell a slightly different story---in all cores, there appears to be a clear (albeit noisy) decrease with depth. To me, this is the trend that really matters, since it speaks to the total OC preservation flux, rather than relative proportions. I point to specific instances where I think this slight shift in focus would improve the clarity of arguments in my line-item comments, below.

Reply: We focus on the fraction of FeR and OC-FeR for several reasons, first, most publications dealing with this topic report these “fraction” data, so this makes it easier to directly compare our findings with other studies. Second, as we mention in Line 86, Fe/Al, Fe, FeR and fFeR are strongly related and show the same downcore pattern and

therefore describing changes in one of the parameter relates also to the other ones and third, we believe that the fOC-Fe relates to the amount of OC sequestered by Fe oxides to TOC, and is thus more suitable than OC-Fe content for comparison among sediments with potentially variable TOC background contents.

But we agree that, as pointed out, there are interesting small (noisy) differences between these parameters, for this reason we show e.g. OC-Fe % in the supplementary figure 7. See our reply and how we aimed our manuscript to the specific points below. Also see our reply to reviewer 1.

The observation that iron-bound OC decays at roughly the same rate as bulk OC is interesting and should be explored more. Can the authors speculate as to some mechanistic reasons why this might be the case? (potentially using their Fe speciation data shown in Fig. 3).

Reply: We agree that the OC-FeR decrease is an interesting observation in regard to discussion about the role of FeR as an effective mechanism for carbon burial. We modified the discussion and now provide some ideas and possibilities as to why we see this change in the OC and OC-Fe content. But we have to admit that without further investigations about the nature of the OC bound to FeR as well as a better knowledge of the exact Fe phases present, this is difficult to answer. Moreover, factors other than diagenetic processes, such as environmental changes, sediment source changes and OC input changes, during the time span investigated here probably play a role. The fact remains that despite strong Fe reduction in the sediments (in all three summers), quite a substantial portion of OC-Fe found at the sediment surface survives this redox process.

We added the following sentences to the discussion part of the revised manuscript (Line 146-162): "Moreover, the gradual decrease of fOC-FeR and OC-FeR (supplementary figure S7) with increasing depth could be related to the domination of more crystalline iron oxide phase of the FeR pool below the surface sediments (Fig. 3). Thus, the loss of OC-FeR may be caused by maturation of these reactive iron phases and an accompanied release of the bond OC40. However, the OC-FeR association not only protects the OC form degradation. The association is also believed to have a stabilising effect on the iron oxides and, therefore, helps to prevent the transformation to more crystalline phases e.g. 41. Furthermore, a large amount of the less reactive iron phases (e.g. goethite and hematite) are allochthonous and OC associated to these phases, probably via mono- or multi-layer sorption, is more accessible for microbial degradation. Further investigations are required to quantify the role of the different Fe(III) phases within the reactive iron pool in stabilising OC in natural sediments. Nevertheless, the decreasing trend of the OC-FeR content is accompanied by a decrease in the total OC content but this trend is much less pronounced in the fOC-FeR profiles. This indicates that, even though the total OC content decreases, a large fraction of total OC content is associated with FeR. It needs to be considered that factors other than diagenetic processes, such as environmental change and OC input changes, probably play a role during the time span investigated here."

We also modified supplementary figure S7 to include the TOC content.

Omission of pore-water O₂ data. Throughout the manuscript, the authors discuss the possibility of FeR precipitation below the oxygenated surface sediments. While the Fe²⁺ profiles are a useful redox indicator, I think it would be very helpful to also include pore-water O₂ profiles (I'm sure these data exist, no?)

Reply: We agree, it would be helpful to have O₂ measurements. However, as we used Rhizons to recover our pore water samples we were unfortunately not able to analyse pore water O₂ profiles at the same samples. We believe that the provided FeR, OC-Fe and pore water Fe measurements are sufficient enough to estimate the depth of the Fe²⁺/Fe³⁺ redox interface and for the investigation of the role of reactive iron on organic carbon stabilisation in marine sediments.

The sentence on L293-295 makes me also wonder if bioturbation could be influencing the FeR content of surface sediments; having pore-water O₂ information could also help speculate on the role (or lack thereof) of bioturbation.

Reply: Whether bioturbation influences the FeR content due to the injection of oxygen is an interesting question. We now cite a recent work from Solan et al 2020 which investigates the bioturbation depth in 2017 and 2018 at the same locations. They show that the mean bioturbation depth is very shallow in the Barents Sea (<1cm). As the iron redox interface was remarkably stable in all investigated cores during our sampling campaigns in summer 2017, 2018 and 2019 we believe that bioturbation had only a minor effect on variable O₂ injection into the sediment. In the revised manuscript we added the following sentence to clarify this (Line 103-106): "Moreover, the stable redox interface also indicates only minor disturbance of the sediment column through bioturbation which is in accordance with a recently reported very shallow mean bioturbation depth (<1 cm) at all investigated stations⁴³."

Use of liquid-decarbonated %OC data. I would like to see a bit of discussion on how the authors think the liquid acidification procedure will bias %OC and, particularly, OC-FeR results. The carbonate content of these sediments was not reported, but presumably if there are considerable carbonates then %OC would be biased upward relative to unacidified samples. On the other hand, some iron phases (e.g., FeS) are known to be dissolved by HCl, which could remove some %OC and thus bias values downward relative to unacidified samples. Was DOC or Fe concentration of the supernatant measured? I suspect the authors could spend a few lines to come to a logical conclusion as to whether the reported values should be treated as minimum estimates, maximum estimates, or neither.

Reply: We did not analyse DOC in the supernatant as it is "contaminated" by the carbon containing solvents (which should be removed by this treatment and the following washing steps) we used for the FeR extraction.

The acidification of the sediments should not significantly affect the OC content, it's a standard analytical procedure to determine the OC content in marine sediment, we accounted for the weight loss during the experiment (supplementary information, equation 1), we clarified this now in the method section in the revised manuscript "To account for the mass loss during the extraction experiment we applied the mass balance calculation of

Salvadó et al. 2 (supplementary information)”. We also added some more information about this in the supplementary information.

All sediment samples have been acidified prior OC analysis and we did not analyse the Fe concentration of the supernatant during the decarbonation process for the OC analysis. The Fe_R extraction should release all OC-Fe prior to the acidification and the acidification should therefore not alter the OC content. But, yes, we cannot exclude that the acidification of the bulk sediment affected the Fe minerals and the OC-Fe bonding in the bulk sediment which may decrease total OC measured in the sediments prior to the extraction experiment.

Minor Comments:

Line 23 (and 127-128): The mention of manganese here seems rather out-of-the-blue and given without context. I would appreciate a sentence or two in the abstract (and, especially, in the introduction) that articulates why Mn should also be included in this study (e.g., what are the open questions? How does this relate to OC-Fe? What has been studied in the past in this regard?) The majority of the manuscript (including the title!) deals specifically with iron, so placing Mn oxides within this context will better justify to the reader why these data are included here.

Reply: We agree and slightly rephrased the abstract to include manganese oxides. We also added the following paragraph in the introduction to clarify why manganese has been included in this study: “Besides Fe(III) phases, Mn(III/IV) (oxyhydr)oxides also strongly interact with OC in marine sediments³⁴⁻³⁶. However, similar to the OC-Fe_R coupling, OC-Mn_R in marine sediment has so far only been investigated in surface sediments and a large deficiency of information remains on the abundance of carbon associated with manganese oxides and their potential role in stabilisation OC over longer timescales. It is therefore unclear if Mn oxides help to transfer OC from the sediment surface carbon cycle to the geological carbon cycle or if Mn_R plays a minor role in OC stabilisation compared to Fe_R^{34,35}.”

Line 43: I’m not sure if Berner 1970 is an appropriate reference here---this paper deals with OC as substrate for sulfate reducers, which leads to sulfide production and eventual pyrite precipitation. While, technically, this is a “association between iron and OC”, I don’t think it quite supports what the authors are articulating here.

Reply: *We rephrased the sentence and omitted the Berner et al. citation.*

Line 52-55: The way this paragraph is set up, the authors are specifically discussing OC-Fe_R that has precipitated at the Fe(II)/Fe(III) redox boundary. This OC has, by definition, already “bypass[ed] the efficient oxic degradation regime...”, given that this redox boundary sits below the oxic zone. I suggest the authors rephrase this paragraph slightly to encompass allochthonous OC-Fe_R, which should indeed behave in the way the authors describe here.

Reply: *We modified the sentence accordingly.*

Line 219: This also results in an accompanying increase of Fe/Al, which should be noted

Reply: This is noted now.

Line 256-257: I don't think this sentence is strictly true. One would predict that an increase in weight % FeR would lead to an increase in weight % OC-FeR, but wouldn't necessarily lead to a higher proportion of total OC bound to iron.

Reply: We modified the sentence accordingly.

Line 259-261: I think this sentence is overstated. When I plot fFeR vs. fOC-FeR, I see a pretty strong correlation ($R^2 = 0.7$) at site B16. But, again, I think it's the weight % FeR vs. weight % OC-FeR correlation that is equally, if not more, important. Interestingly, when I plot these variables against one other, I see a relative strong correlation at sites B15 ($R^2 = 0.48$) and B16 ($R^2 = 0.85$), but not at sites B13 and B14. This points to some nuanced differences between sites that I think could be explored and discussed more.

Reply: We modified the sentence to lessen the strength of our statement. However, looking at the fFeR (blue line) and fOC-Fe (yellow line) profiles at figure 2 does not provide evidence that they are related. Moreover, we agree that there might be some kind of correlation at station B16 and maybe B15, however looking at the correlation (OC-Fe vs. FeR) for B16 for example it becomes clear that the r^2 value is biased by four points and is therefore not very reliable:

We agree that there is "nuanced differences between the sites" which we haven't discussed, but we believe that our manuscript is already very dense and compact, and we do not believe the difference is significant enough to alter the main message of the manuscript.

Line 312-313: Again, I don't think this sentence is strictly true. Is the increasing OC:Fe ratio deeper in the sediments at sites B14 and B15 really evidence for coprecipitation? Alternatively, this could just be subtle differences in the rate of decrease of iron and OC with depth. When I look at reactive iron weight % with depth, it appears to be dropping across the two regions where OC:Fe is increasing (particularly at ~15cm at site B15, i.e., at the base of the red layer). To me, this argues against co-precipitation but rather suggests

OC decreases with depth at a slower rate than Fe in this region. This should be discussed in more detail.

Reply: We agree that this is a difficult assumption and we modified the sentence to clarify its weakness. We like to point out that we discuss the problems and uncertainties related to the OC:Fe values at line 199-212 and mention e.g. at line 207: "The molar ratio of OC:Fe might therefore be biased and especially low OC:Fe ratios, as in core B15 and B16, should be interpreted with care".

Fig 3: I think there is an error in either Fig. 3 or Table S3---FeOx-2 and FeOx-3 (called Fe-S2 and Fe-S3 in Table S3 for some reason?) appear to be flipped.

Reply: Thanks for pointing this out. This is now fixed.

All other very minor issues and technical comments highlighted by reviewer 2 have been considered in the revised version.

Reviewer #3

Main comments

The discussion on the role Fe-oxides comes across as one sided as there is a lack of discussion on the role of clay minerals and the surface area that they provide for stabilizing sedimentary organic carbon (e.g., Keil et al., 1994; Mayer et al., 1994). What do the early diagenetic TOC decreases mean for downcore organic carbon - bulk mineral (surface area) relationships? I would find the mineral surface area discussion of great value as do Fe-oxides stabilize a disproportionate amount of OC relative to silicates and other mineral classes based on their abundance? I do find it remarkable that reactive Fe, which comprises less than 2 wt.% in the "normal" background sediments, should participate in stabilizing 20% of the OC, while silicates which are likely present in double digit quantities (?) would only be stabilizing part of the remainder of sedimentary OC. What kind of specific surface areas can we expect from these reactive Fe phases? Can their theoretical mineral surface areas explain the disproportionate amount of OC stabilized by them (perhaps there is measured surface area data available)?

Reply: We completely agree that the role of clay minerals and the surface area they provide are interesting and important factors for the stabilisation of organic carbon in marine sediments. In the revised manuscript, we now mention in the introduction section that clay minerals also play an important role in OC stabilisation in sediments. Moreover, we agree that it is now necessary to investigate the interrelations between these factors (role of clay minerals and surface area) and metal oxides to better understand the controlling mechanisms of OC storage in marine sediments. However, we believe that this is beyond the scope of this manuscript. We show for the first time that OC-FeR is stable over long timescales and that MnR does not play a role in enhancing carbon burial in marine sediments. This is a great step forward in understanding the fate

of OC in marine sediments and we can now build on these new findings and see how they relate e.g. to the role of clay minerals.

Yes, it is indeed remarkable that FeR seems to stabilize such large fractions of the TOC pool which is why our study about the fate of the OC-Fe association over long timescales is so important.

Unfortunately we did not measure the mineral surface area and we believe that a theoretical mineral surface area assumption would be more guessing than knowing and would therefore not be helpful. As we mentioned before, here we show for the first time that FeR minerals play an important role in the OC storage, their many questions which arise from this finding which we can only answer with further investigations.

At the same time, why is the amount of OC stabilized by Fe-oxides insensitive to the amount of reactive Fe available (e.g., lines 257-259; lines 276-277)? In some way, this contradicts OC-MSA relationships (e.g., Keil et al., 1994; Mayer et al., 1994) seen for bulk sediment. In an additional twist to this discussion, what are the relationships between silicates and Fe-oxides and are we able to tease apart OC associations between one or the other? See for example figure 2 in Kleber et al. (2007).

Reply: Our data indicate that FeR is not the controlling factor for the amount of OC bound to FeR, we discuss these findings and possible reasons (e.g. line 162-167) but further investigations are required to fully understand this new observations (line 251-254).

We are sorry but we cannot answer this question with our investigation and we believe that this is beyond the scope of this manuscript. In the presented manuscript we focus and on the importance of reactive iron phases for OC burial in marine sediments.

How was fOC-FeR calculated? The methods in this manuscript mainly provide a distillation of the wet chemical approach. Adding explicit equations in a supplementary file would be useful (beyond referencing Lalonde et al., 2012, which are light on these details). For the sake of formality, adding equations for the other calculated parameters (e.g., FeR) would also be useful to see how blanks were subtracted, etc.

Reply: For our analytical approach we refer to Lalonde et al 2012, which is indeed light on these details, and we therefore also refer to the follow up study from Salvado et al. 2015 (line 306) which provides a detailed description of the analytical procedure. The equation from Salvado et al for the calculation of OC-FeR was initially provided in the supplementary table S3. In the revised manuscript we moved the equation into the supplementary information and added some clarifying notes about the calculation of fFe-OC, FeR and all other parameters shown.

Precipitation and coprecipitation of organic carbon (with iron oxides) is often discussed in the manuscript and here I wonder what does this mean? With “precipitation” a process of transitioning from dissolved to solid phase is suggested/implied. Is dissolved organic matter removed from porewater and coprecipitated on authigenic Fe-oxides? Does dissolved organic matter solubility change across oxic-anoxic transition zones? Is there

enough pore water dissolved organic matter present to support such process (e.g., lines 50, 296, 314)?

Reply: We describe the process of Fe(III) precipitation and the accompanied coprecipitation or adsorption of OC at (Line 46-51). The exact process of how this binding works and which type of organic matter is bound to FeR is currently investigated and discussed in the literature. We like to refer to the references given in our manuscript as we think that a discussion about the physical and chemical details on a molecular basis is beyond the scope of this manuscript.

A local discussion which I think is worth having concerns Svalbard, which acts as a point source of petrogenic organic carbon (reworked kerogen) to the adjacent sediments (Kim et al., 2011). The recalcitrance of petrogenic organic carbon and its behavior is quite different from that of freshly synthesized organic carbon (e.g., from soils or from the ocean) and its associations with minerals (e.g., Blattmann et al., 2019). What role could rock-derived Fe and kerogen be playing in these sediments? Could they be one reason for the insensitivity of reactive Fe abundance to Fe-OC?

Reply: We can only speculate about this as we do not have any data about the composition of the organic matter. This would be indeed very interesting and should be investigated in the next step. It would also be very interesting to sample the soil and the river streams in Svalbard to identify how much OC-Fe enters the ocean from Svalbard and to discover how stable it is during transport from its land source until burial in the sediments. We raise this question of the “insensitivity of reactive Fe abundance to Fe-OC” in our manuscript (line 162-167) and discuss this in the paragraph below these lines.

The speculative discussions involving manganese (minor element with much less than 1 percent abundance) and arsenic (trace element) seem too long for this type of article. Why should such minor and trace elements have a detectable effect (using the bulk and semi-bulk methods used in this study) on sedimentary OC stabilization? Beyond competing with OC sorption, multivalent ions can also be involved in cation-bridging (Keil and Mayer, 2014), so in my opinion the effect could also go the other way. I suggest keeping a hypothesis-driven focus for the discussion.

Reply: We would ask “why should Mn not have a large influence on OC stabilisation in relation to its concentration?”. It is well known that MnR has (as FeR) a strong affinity to OC in surface sediments, but its “effect on carbon stabilisation in natural sediments is almost completely unconstrained (line 222)”, therefore we believe that it is important to study the fate of OC-MnR on longer timescales. To clarify this we added a paragraph in the introduction (line 68-74). We also slightly shortened the “Mn paragraph” (line 219-231) as we agree that it is too long for this type of article.

Finally, how are the conclusions to be placed in the global context with so much variability as reported by Lalonde et al., 2012? In this regard, I think it would be useful to contextualize and include (wherever possible) (estimates of) parameters in the discussion such as mixed layer depth, sedimentation rate, Fe and Mn porewater (e.g., perhaps

combine into one figure in supplemental like Froehlich et al., 1979), and oxygen exposure time of organic carbon (e.g., Hartnett et al., 1998).

Reply: This is a good idea, and this is also what we are heading for, to provide a detailed picture of the processes in marine sediment affecting OC burial. Unfortunately, we are not there yet and our work is one important piece on the way to this goal. We need further investigation to create a figure equivalent to that in Froehlich et al 1979.

Minor Comments:

Lines 114-122: This goal, while interesting and of course highly relevant for Earth Science, in my opinion, somewhat detracts from the main scope and message that the title and abstract outlines. I would suggest moving this into the later discussion or “implications and conclusions” section. This is merely a suggestion.

Reply: We moved these sentences to the implication and conclusion section.

Lines 123-124: I suggest deemphasizing/removing the “focus on the potential effect of variable iron fluxes...” as this effect is discussed extensively later and deconvolved from the effects of most pertinent interest here.

Reply: We removed the mentioned part of the sentence.

Line 43: Soil scientists were much earlier to recognize this and in my opinion is worth pointing out if such historical references are made. See review by Beutelspacher (1955).

Reply: We rephrased the sentence accordingly and now cite Beutelspacher (1955).

Lines 194-199: Was organic carbon content corrected to weight loss of sample due to loss of acid-soluble minerals?

Reply: Yes, we mentioned this in line 318 and we now add additional information in the method section and in the supplementary information to clarify this.

Line 208: The introduction of the acronym fOC-FeR is somewhat confusing; here, it is not clear how it is defined whether as an “effect of xx on yy” or as what I think it is.

Reply: We modified the sentence for clarification. We also added additional information in the supplementary information to clarify this.

Fig. 2: I suggest uniform scaling and removing porewater Fe average (moving this to the supplemental) and simply indicating the oxic-anoxic transition zone with a colored band within the profile to make the figure less busy.

Reply: We changed the figure accordingly; the scaling is now uniform and we removed the grey Fe average. We did not include a coloured band to indicate the oxic-anoxic transition zone, because we don't know the exact oxygen penetration depth. We also found that this coloured band would interfere with the grey area at station B15 and would not help to reduce the busyness of the figure.

Lines 271-273: Needs reference(s).

Reply: Reference has been added.

Line 300: Please add references for who previously assumed this.

Reply: Reference has been added.

Line 306: Don't the blue bars show the sum of FeR from the sequential extractions? Perhaps I am confused here.

Reply: There was a mistake in the figure caption, this is now solved.

Lines 309-311: This sentence seems redundant.

Reply: We deleted the sentence.

Lines 324-326: There is a problem in the logic of this sentence: how can coprecipitated reactive iron be more reactive towards microbial reduction (this is a process)?

Reply: We modified the sentence for clarification.

Line 328: What does preferentially mean in this context? Preferentially in what way?

Reply: We modified the sentence for clarification.

Line 332: What is meant with impure?

Reply: We modified the sentence for clarification.

All other very minor issues and technical comments highlighted by reviewer 3 have been considered in the revised version.

REVIEWER COMMENTS

Reviewer #1 (Remarks to the Author):

Faust and coauthors have provided thoughtful and thorough responses to my initial comments, and I believe the manuscript is much stronger following those changes and the changes recommended by the fellow reviewers. I have a few minor comments based on those changes and feel the paper is ready for publication once these are addressed.

Line 152: "bond" should be "bound"

Line 153: "form" should be "from"

Line 155 - 158: Please provide a reference for the statement on degradability of OC relative to FeR phase.

Line 205: This statement now slightly conflicts with your previous statement and response to reviewers (lines 155-158 in the revised manuscript). You will need to specify that you (or more precisely the cited reference) are comparing coprecip and adsorption for the same phase of iron here and you are comparing iron phases above.

Michael Shields (mshields@tamu.edu)

Reviewer #2 (Remarks to the Author):

Synopsis

The revised version of this manuscript exhibits significant improvements in clarity and nuance relative to the initial submission. However, I feel that there are still areas in which further revisions are warranted or where the authors' response to initial reviewer comments was not entirely satisfactory. I articulate these points in detail below, followed by a list of line-item comments. Please do not hesitate to contact me with any questions regarding this review.

Sincerely,

Jordon Hemingway

+1 760 445-3714

jordon_hemingway@fas.harvard.edu

Larger points

Framing of the question (Abstract & Introduction)

In my reading, this manuscript provides great insight for two important and currently outstanding questions regarding iron-bound OC: (i) how much OC-Fer is derived by allochthonous vs. authigenic processes? and (ii) at what rate is OC-Fer remineralized below the oxic zone in shelf sediments? However, in the current form, and largely summarizing previous publications on this topic, the introduction largely ignores question (i) and *presumes* that authigenic precipitation of Fe(III) phases leads to OC-Fer enrichment in surface sediments (e.g., beginning on L42). However, the authors (rightfully) conclude that this is likely not the case. I therefore suggest tweaking the introduction to properly frame the question as: "What is the *source* and *fate* of iron-bound OC in shelf sediments?" This will likely be an easy change to make, and I think it will help

in distilling the contributions of this manuscript. (For example, the role of allochthonous vs. authigenic phases is not currently mentioned in the four main points to be addressed on L80-85.)

Study-site details

Somewhat echoing Reviewer #3, I would like to see some discussion (e.g., on L80) related to this particular study site. For example, why the Barents Sea shelf area was chosen for this study? What is the global importance of this site? How is this location ideally suited to address both the source and fate of iron-bound OC? Without these details, the motivation seems somewhat arbitrary. Again, this will likely be an easy fix to make (there are already some details in the caption of Fig. 1 and in the discussion), and I think it will again help to distill the motivation and importance of the current study, in addition to contextualizing these results when the authors compare to previous studies later in the discussion.

Absolute vs. fractional iron-bound OC content

While I recognize the utility of using fOC-Fer when comparing results between sites, across studies, etc. (e.g., L53-55)—particularly since differences in total “background” OC content could mask iron-bound OC trends—I still feel strongly that it makes little sense when assessing

downcore processes and mechanisms and that it leads to false or misleading statements. For example, the following sentence is quite an overstatement and is not justified (similarly L274-277): “...sedimentary organic carbon is associated with reactive iron and shielded against mineralization back to CO₂ on at least millennial timescales” (L27-28). Looking at the absolute amount of OC bound to Fe_R (termed “OC bond to Fe_R (%)” in the SI data table), I calculate a down-core decrease of 95% (!), 64%, 74%, and 76% in cores B13, B14, B15, and B16, respectively (see attached plot)---clearly, a majority of Fe-bound OC *is* remineralized over millennial timescales, even if the proportion of total OC remains relatively constant! (Of course, as the authors allude to, it remains possible that secular increases in OC input could describe some of these downcore trends, but I don’t think the authors are arguing this to be the case.)

Put differently, one could presumably observe down-core variability in the fraction of total OC that is bound to Fe_R *as a result of processes that are completely independent from iron*; For example, if more or less OC becomes bound to clay minerals, selectively respired, etc. Conversely, one could (and does) observe a relatively stable fraction of OC that is bound to Fe_R despite the fact that the absolute content decreases markedly. By consistently normalizing to total OC content, this manuscript risks conflating processes that are *mechanistically* tied to OC-Fe interactions with processes that are independent but result in a *statistical* relationship between OC and Fe. (Again, somewhat echoing Reviewer #3, these differences could be articulated with some more discussion of iron-bound OC within the context of other hypothesized OC preservation mechanisms.)

The authors additionally state in their response that: “...as we mention in Line 86, Fe/Al, Fe, Fe_R and fFe_R are strongly related and show the same downcore pattern...” This is true (although note that total weight % Fe does not correlate with at site B13) but irrelevant! None of these metrics

shows a consistent correlation with either weight % OC-Fe_R or fractional OC-Fe_R across all study sites. This result could in fact be interpreted as support for one of the authors' main conclusions---that authigenic precipitation of Fe(III) phases within the redox gradient does not necessarily lead to enhanced iron-bound OC content. Furthermore, a plot such as the one above (combined with the relative stability of fOC-Fe_R) supports the authors' other main conclusion---that iron-bound OC is not all released back into solution immediately below the redox gradient; rather, iron-bound OC persists and is respired at a similar rate as bulk OC.

OC content measurements

The response to this point was not satisfactory (note that this issue was also raised by Reviewer #3). The authors' response appears to concern *mass loss during reactive iron extractions* (e.g., L13-15 of the SI notes: "Additionally, to account for the mass loss during the extraction experiment we applied the mass balance calculation of Salvado et al. 2. Equation 1 was used to determine %OC-Fe_R..."). Strictly speaking, this mass balance is correct. However, my original comment (and I assume that of Reviewer #3) concerns *mass loss during acid decarbonation*, not during reactive iron extractions. The authors have yet to address this issue, despite its potential importance.

Specifically, liquid-HCl decarbonation of the "initial" samples (using the nomenclature from SI notes Eq. 1) will lead to loss of reactive iron phases in addition to carbonates, thus liberating some iron-bound OC and biasing "OC_{initial}" to lower values. Quoting from the cited Poulton and Canfield (2005) paper: "The HCl extraction removes a variety of Fe phases, including Fe (oxyhydr)oxides such as ferrihydrite, lepidocrocite, goethite and hematite, and some Fe from sheet silicates." (pp. 210; noting however that they are referring to 12N HCl rather than the 1.2N used here). If this loss is significant for the current sample set, then I would expect resulting OC-Fe_R % values to be artificially biased upward due to a downward bias in the denominator of Eq 1 in the SI notes. Furthermore, even if these biases are small in terms of total OC content, they may become significantly larger when propagated onto Fe-bound OC, as this represents only a fraction of total OC.

In their response, the authors state that "[liquid-HCl decarbonation is] a standard analytical procedure to determine the OC content in marine sediment..." While this may be true, that does not mean that the technique is free of bias, as described in detail in, for example, the following publications:

Bao et al. (2019) *Radiocarbon*, **61**, 395-413.

Brodie et al. (2011) *Chemical Geology*, **282**, 67-83.

Komada et al. (2008) *Limnology & Oceanography: Methods*, **6**, 254-262.

Minor comments

L39: "Fe_R" has not yet been defined as "reactive iron"

L44: Here Fe²⁺ is attributed specifically to "dissimilatory iron reduction" but below (L64) it is attributed to "reductive dissolution of Fe_R through biotic and/or abiotic processes." This should be updated for consistency.

L89: It is unclear from the text that “Fer content” specifically refers to the weight % of reactive iron in sediments.

L92: There appears to be a typo / formatting issue: “e.g.²⁹, i.e.,...”

L94: “fFer” is not yet defined.

L140: I’m slightly confused by the line: “...extraction of *none* or less-reactive iron phases...”

L150: I believe the authors mean “predominance” rather than “domination”

L152: Change “bond” to “bound”

L155-158: What is the evidence for this statement? How do the authors know that a large fraction of goethite and hematite are likely allochthonous?

L172: The authors need to clearly articulate that in this manuscript “OC:Fe (molar ratio)” specifically refers to the molar ratio of iron-bound OC to reactive iron phases (not total OC to total iron).

L186-189: In my opinion, this is the clearest articulation of one of the main conclusions of this study! (see my “Framing the question” point, above).

L201: Remove comma after “For,”

L217-220: Is it really true that the “...strong relationship between arsenic and Fer implies that arsenic sorption changes the mineral surface properties and reactivities of the Fe(III) phases and, therefore, their capacity to bind to OC.”? Alternatively, this could simply result from an affiliation between As and Fe, with zero implications for OC binding. This sentence seems very speculative.

L226: “MnR” is not yet defined.

Fig. 3: There still appears to be an inconsistency between FeOx-1, FeOx-2, and FeOx-3 content as shown in the figure vs. that reported in the SI Table. For example, the figure shows FeOx-3 at site B13 consistently around 0.30%, whereas this drops to 0.05% at 9.5cm in the SI Table.

Reviewer #3 (Remarks to the Author):

Dear Faust et al.,

Your work “Millennial scale persistence of organic carbon bound to iron in Arctic marine sediments” has been revised extensively and the bar has been raised thanks to your efforts. Several typos stood out to me and I urge a careful check prior to submitting a final version. Here are the ones I found by line number:

72: been

74: stabilising

100-105: This sentence is too long. I recommend reorganizing.

109: stations

150: ...domination of more crystalline iron oxide phases in the FeR pool...

152: bound

153: from

176: suggested comma after sediments

178: stations

192-193: This sentence is a fragment.

195: station lowercase

201: comma after for

255: mechanisms

Generally, doublecheck the usage of the words bound, bind, bounding, binding, etc., which I am also not entirely sure about (see e.g., line 26).

Please doublecheck the formula in the supplementary information. In lines 23 and 24, the masses are both before the extraction, which according to the methods should both be equal to 0.25 grams. Shouldn't there be a mass after extraction to account for mass loss?

In your response to the role of manganese in stabilizing OC, I would say there are in my opinion probably larger factors than manganese (e.g., oxygen exposure time, mineral surface area, etc.), which co-determine the amount of OC preserved in sediments on a bulk level, but I think including this (I think provocative idea) is good for science and should stimulate new discussions and research directions. This is a meticulously executed study, which advances our basic understanding on the role of iron in the organic carbon cycle and I recommend accepting this work with minor revisions.

Sincerely,

Thomas Blattmann

20.10.2020 Yokosuka

Note: All line numbers in the reply sections relate to the revised manuscript.

Reviewer #1:

Minor Comments:

Faust and coauthors have provided thoughtful and thorough responses to my initial comments, and I believe the manuscript is much stronger following those changes and the changes recommended by the fellow reviewers. I have a few minor comments based on those changes and feel the paper is ready for publication once these are addressed.

Line 155 - 158: Please provide a reference for the statement on degradability of OC relative to FeR phase.

Reply: We now provide a reference for this statement.

Line 205: This statement now slightly conflicts with your previous statement and response to reviewers (lines 155-158 in the revised manuscript). You will need to specify that you (or more precisely the cited reference) are comparing coprecip and adsorption for the same phase of iron here and you are comparing iron phases above.

Reply: Yes, we agree that there is a conflict with the previous statement and we modified the sentence in line 205 (now 223) for clarification.

All other very minor issues and technical comments highlighted by reviewer 1 have been considered in the revised version.

Reviewer #2

Main comments

Framing of the question (Abstract & Introduction): In my reading, this manuscript provides great insight for two important and currently outstanding questions regarding iron-bound OC: (i) how much OC-FeR is derived by allochthonous vs. authigenic processes? and (ii) at what rate is OC-FeR remineralized below the oxic zone in shelf sediments? However, in the current form, and largely summarizing previous publications on this topic, the introduction largely ignores question (i) and presumes that authigenic precipitation of Fe(III) phases leads to OC-FeR enrichment in surface sediments (e.g., beginning on L42). However, the authors (rightfully) conclude that this is likely not the case. I therefore suggest tweaking the introduction to properly frame the question as: "What is the source and fate of ironbound OC in shelf sediments?" This will likely be an easy change to make, and I think it will help in distilling the contributions of this manuscript. (For example, the role of allochthonous vs. authigenic phases is not currently mentioned in the four main points to be addressed on L80-85.)

Reply: Yes, we were, and still are, cautious with any statement about the source of the reactive iron-bound organic carbon (OC-FeR). Our combined pore water and solid phase data indicate that significant amounts of OC-FeR are allochthonous (i.e., inferred from the fact that not all of it is authigenic), but we cannot calculate “how much” OC-FeR is allochthonous. As highlighted in our introduction, the most recent scientific consensus is that authigenic coprecipitation is not the dominant mechanism facilitating the carbon-iron bounding in marine sediments. We agree with your comments and are grateful for the support to emphasize this topic.

We have now modified the abstract to emphasize our proposal that large amounts of OC-FeR are allochthonous (Line 25-28). We also extended the “four main points” (Line 98), and added the following sentence to the introduction (Line 82-93): “Iron and manganese (oxyhydr)oxide reduction plays an important role in organic matter degradation in this region^{29,30} and it is therefore, a suitable location to study the combined diagenetic fate of OC and iron and manganese. Moreover, downcore investigations of OC-FeR will not only provide a better understanding of the role of early diagenesis in OC-FeR generation and stability; they will also help to reveal the source of the OC-FeR (allochthonous versus autochthonous), and allow to identify the relative contributions of OC-FeR that was formed on land, during the transport process, or at the sediment-water interface.”

Study-site details: Somewhat echoing Reviewer #3, I would like to see some discussion (e.g., on L80) related to this particular study site. For example, why the Barents Sea shelf area was chosen for this study? What is the global importance of this site? How is this location ideally suited to address both the source and fate of iron-bound OC? Without these details, the motivation seems somewhat arbitrary. Again, this will likely be an easy fix to make (there are already some details in the caption of Fig. 1 and in the discussion), and I think it will again help to distill the motivation and importance of the current study, in addition to contextualizing these results when the authors compare to previous studies later in the discussion.

Reply: We would like to note here that in the first submission of our manuscript, the introduction included a discussion about the study area. Reviewer #3 suggested to move this part into the conclusion or discussion part, which we did. Thus, prior to the second review round, the first paragraph of the “Synthesis and implications” section was part of the introduction.

We have now moved this paragraph back into the introduction section (Line 85-93).

Moreover, we have added two more references (29 and 30) and an additional sentence to clarify why this region is a suitable location to investigate the source and fate of iron-bound OC (Line 84).

Absolute vs. fractional iron-bound OC content: While I recognize the utility of using fOC-FeR when comparing results between sites, across studies, etc. (e.g., L53-55)—particularly since differences in total “background” OC content could mask iron-bound OC trends—I still feel strongly that it makes little sense when assessing 2 downcore processes and

mechanisms and that it leads to false or misleading statements. For example, the following sentence is quite an overstatement and is not justified (similarly L274- 277): “...sedimentary organic carbon is associated with reactive iron and shielded against mineralization back to CO₂ on at least millennial timescales” (L27-28).

Reply: We understand that this statement can be seen as an overstatement due to the gradual downcore-decreasing trend of OC-Fe and fOC-Fe. We have therefore deleted or modified the sentences in lines 27-28 and 280-283 accordingly.

Looking at the absolute amount of OC bound to Fe_R (termed “OC bond to Fe_R (%)” in the SI data table), I calculate a down-core decrease of 95% (!), 64%, 74%, and 76% in cores B13, B14, B15, and B16, respectively (see attached plot)---clearly, a majority of Fe-bound OC is remineralized over millennial timescales, even if the proportion of total OC remains relatively constant! (Of course, as the authors allude to, it remains possible that secular increases in OC input could describe some of these downcore trends, but I don’t think the authors are arguing this to be the case.)

Reply: We agree that we need to address this decreasing trend of OC-Fe more clearly. We have therefore modified the results and discussion part (Line 170-178) to point out that gradual remineralization of iron-bound OC may occur over time. However, we also caution that the calculated values of downcore OC-Fe_R decrease strongly depend on the core section investigated, and using the top maximum and bottom minimum values for such a calculation is an oversimplification of the data set. For example, sediment cores B15 and B16 show stable OC-Fe_R values over time in the lower half of each core. Moreover, by plotting fOC-Fe_R versus OC-Fe_R it becomes clear that both parameters show the same decrease/pattern for all of the cores:

Nevertheless, we agree that calling the downward decrease of fOC-Fe_R less pronounced than the OC-Fe_R results was incorrect, and we have changed our manuscript accordingly. The figure above has now been added to the supplementary information (Fig. S7).

Put differently, one could presumably observe down-core variability in the fraction of total OC that is bound to FeR as a result of processes that are completely independent from iron; For example, if more or less OC becomes bound to clay minerals, selectively respired, etc. Conversely, one could (and does) observe a relatively stable fraction of OC that is bound to FeR despite the fact that the absolute content decreases markedly. By consistently normalizing to total OC content, this manuscript risks conflating processes that are mechanistically tied to OC-Fe interactions with processes that are independent but result in a statistical relationship between OC and Fe. (Again, somewhat echoing Reviewer #3, these differences could be articulated with some more discussion of iron-bound OC within the context of other hypothesized OC preservation mechanisms.)

Reply: We now address this issue and clarify that the downcore fraction of organic carbon bound to reactive iron (fOC-FeR) can be affected by factors that are completely independent from iron. We have added and adapted the following sentences (Line 172-183):

“Nevertheless, the decreasing trends of OC-FeR and fOC-FeR are accompanied by overall declining total sedimentary OC content with increasing depth at all stations (Fig. 4), and we cannot rule out that downcore variability in the fOC-FeR has been affected by processes completely independent from iron. In fact, we fully acknowledge that the downcore patterns in the absolute amounts of OC bound to FeR may have been affected by various processes. These include the remineralization of iron bound OC over time, but also a combination of chemical, physical and biological processes that affect sedimentary OC records, including a variable fraction of OC being bound to clay minerals or variable amounts of non-bound OC being degraded (reviews by Arndt et al., 2013; LaRowe et al., 2020). Nonetheless, the fact that on average 19.2% of the total organic carbon remains bound to FeR below the oxygenated surface sediment layer still highlights the important role that this OC-FeR association plays in long-term carbon storage, despite the variance in environmental parameters over time.”

The authors additionally state in their response that: “...as we mention in Line 86, Fe/Al, Fe, FeR and fFeR are strongly related and show the same downcore pattern...” This is true (although note that total weight % Fe does not correlate with at site B13) but irrelevant! None of these metrics shows a consistent connection with either weight % OC-FeR or fractional OC-FeR across all study sites. This result could in fact be interpreted as support for one of the authors’ main conclusions---that authigenic precipitation of Fe(III) phases within the redox gradient does not necessarily lead to enhanced iron-bound OC content.

Reply: As suggested we now use this finding to strengthen our main conclusion that authigenic precipitation of Fe(III) phases within the redox gradient does not necessarily lead to enhanced iron-bound OC content (Line 183-187).

Furthermore, a plot such as the one above (combined with the relative stability of fOC-FeR) supports the authors’ other main conclusion--- that iron-bound OC is not all released back into solution immediately below the redox gradient; rather, iron-bound OC persists and is respired at a similar rate as bulk OC.

Reply: Thanks for this thoughtful and constructive suggestion. We believe that our existing Figure 2 already supports the main conclusion that “iron-bound OC is not all released back into solution immediately below the redox gradient” adequately. A plot showing the OC-FeR versus fOC-FeR was already provided in the previous version of this manuscript, supplementary Figure S7. In the revised version of the manuscript, we have now moved Figure S7 into the main text (new Figure 4).

OC content measurements: The response to this point was not satisfactory (note that this issue was also raised by Reviewer #3). The authors’ response appears to concern mass loss during reactive iron extractions (e.g., L13-15 of the SI notes: “Additionally, to account for the mass loss during the extraction experiment we applied the mass balance calculation of Salvado et al. 2. Equation 1 was used to determine %OC-FeR...”). Strictly speaking, this mass balance is correct. However, my original comment (and I assume that of Reviewer #3) concerns mass loss during acid decarbonation, not during reactive iron extractions. The authors have yet to address this issue, despite its potential importance. Specifically, liquid-HCl decarbonation of the “initial” samples (using the nomenclature from SI notes Eq. 1) will lead to loss of reactive iron phases in addition to carbonates, thus liberating some iron-bound OC and biasing “OC_{initial}” to lower values. Quoting from the cited Poulton and Canfield (2005) paper: “The HCl extraction removes a variety of Fe phases, including Fe (oxyhydr)oxides such as ferrihydrite, lepidocrocite, goethite and hematite, and some Fe from sheet silicates.” (pp. 210; noting however that they are referring to 12N HCl rather than the 1.2N used here). If this loss is significant for the current sample set, then I would expect resulting OCFer % values to be artificially biased upward due to a downward bias in the denominator of Eq 1 in the SI notes. Furthermore, even if these biases are small in terms of total OC content, they may become significantly larger when propagated onto Fe-bound OC, as this represents only a fraction of total OC. In their response, the authors state that “[liquid-HCl decarbonation is] a standard analytical procedure to determine the OC content in marine sediment...” While this may be true, that does not mean that the technique is free of bias, as described in detail in, for example, the following publications:
Bao et al. (2019) *Radiocarbon*, 61, 395-413.
Brodie et al. (2011) *Chemical Geology*, 282, 67-83.
Komada et al. (2008) *Limnology & Oceanography: Methods*, 6, 254-262.

Reply: Thanks for this explicit and well-articulated comments. Having conducted this study following the most up-to-date extraction schemes for OC-Fe, we realise there is room for methodological improvement (e.g., acid fumigation instead of liquid acid decarbonation). In fact, any kind of sample pre-treatment will most likely affect certain characteristics of the sample. However, formulating improved methods for OC-Fe extraction from natural samples, and testing their advantages against the existing method, is beyond the scope of this manuscript. In addition, by following existing extraction schemes, our results are directly comparable to published data compilations and allow us to put our findings into a wider scientific context.

To make the reader aware of potential effects of liquid HCl decarbonation on OC-FeR extraction, we have now added the following sentences in the methods section (Line

358-361): “Note that liquid-HCl decarbonation of the bulk sediment samples may also dissolve reactive iron phases in addition to carbonates. This could potentially liberate some iron-bound OC, which would bias our bulk organic carbon results to lower values and thus bias our OC-Fe_R results upwards (supplementary information).” And in the supplementary note, we added the following sentence: “Note that liquid-HCl decarbonation of the bulk sediment samples may also dissolve reactive iron phases in addition to carbonates, which potentially could liberate some iron-bound OC, which would bias our bulk organic carbon results to lower values. Our reported OC-Fe_R values might therefore be biased towards slightly higher values due to a downward bias in the denominator of Eq. 1.”

Minor comments

L155-158: What is the evidence for this statement? How do the authors know that a large fraction of goethite and hematite are likely allochthonous?

Reply: We have no evidence, it is merely an educated guess. We have therefore omitted the statement that we think that large fractions of the less reactive Fe phases are allochthonous.

L217-220: Is it really true that the “...strong relationship between arsenic and Fe_R implies that arsenic sorption changes the mineral surface properties and reactivities of the Fe(III) phases and, therefore, their capacity to bind to OC.”? Alternatively, this could simply result from an affiliation between As and Fe, with zero implications for OC binding. This sentence seems very speculative.

Reply: That is true, we modified the sentence to clarify that we speculate here to highlight the issue and hopefully inspire further research into competitive adsorption of OC and metals onto Fe_R surfaces.

Fig. 3: There still appears to be an inconsistency between FeOx-1, FeOx-2, and FeOx-3 content as shown in the figure vs. that reported in the SI Table. For example, the figure shows FeOx-3 at site B13 consistently around 0.30%, whereas this drops to 0.05% at 9.5cm in the SI Table.

Reply: Thanks for pointing this out, there was a mistake in the column labelling in SI Table.

All other very minor issues and technical comments highlighted by Reviewer #2 are gratefully acknowledged and have been considered in the revised version.

Reviewer #3

Main comments

Your work “Millennial scale persistence of organic carbon bound to iron in Arctic marine sediments” has been revised extensively and the bar has been raised thanks to your efforts. Several typos stood out to me and I urge a careful check prior to submitting a final version.

Reply: All minor issues and technical comments highlighted have been considered in the revised version of our manuscript.

Please doublecheck the formula in the supplementary information. In lines 23 and 24, the masses are both before the extraction, which according to the methods should both be equal to 0.25 grams. Shouldn't there be a mass after extraction to account for mass loss?

Reply: That is correct - they should both be equal to 0.25 g, but there are very small differences (e.g., control 0.2489 g and extraction 0.2505 g). Therefore, this part of the equation deals with the small differences in the initial weights of the duplicates. The percentage of carbon always relates back to 0.25 g (also after the experiment).

In your response to the role of manganese in stabilizing OC, I would say there are in my opinion probably larger factors than manganese (e.g., oxygen exposure time, mineral surface area, etc.), which codetermine the amount of OC preserved in sediments on a bulk level, but I think including this (I think provocative idea) is good for science and should simulate new discussions and research directions. This is a meticulously executed study, which advances our basic understanding on the role of iron in the organic carbon cycle and I recommend accepting this work with minor revisions.

Reply: Thank you very much for these supporting and motivating statements.

REVIEWERS' COMMENTS

Reviewer #2 (Remarks to the Author):

The authors have largely addressed my previous comments and suggestions; I therefore support publication of this manuscript in its current form.

Sincerely,
Jordon Hemingway

Reviewer #2:

The authors have largely addressed my previous comments and suggestions; I therefore support publication of this manuscript in its current form.

**Sincerely,
Jordon Hemingway**

Reply: Thank you for the supportive and constructive reviews which were very helpful to improve our manuscript.